# UNDERSTANDING TOOL-INTEGRATED REASONING

## ABSTRACT

We study why Tool-Integrated Reasoning (TIR) makes Large Language Models (LLMs) more capable. While LLMs integrated with tools like Python code interpreters show great promise, a principled theory explaining why this paradigm is effective has been missing. This work provides the first formal proof that TIR fundamentally expands an LLM's capabilities. We demonstrate that tools enable a strict expansion of the model's empirical and feasible support, breaking the capability ceiling of pure-text models by unlocking problem-solving strategies that are otherwise impossible or intractably verbose. To guide model behavior without compromising training stability and performance, we also introduce Advantage Shaping Policy Optimization (ASPO), a novel algorithm that directly modifies the advantage function to guide the policy behavior. We conduct comprehensive experiments on challenging mathematical benchmarks, leveraging a Python interpreter as the external tool. Our results show that the TIR model decisively outperforms its pure-text counterpart on the pass@$k$ metric. Crucially, this advantage is not confined to computationally-intensive problems but extends to those requiring significant abstract insight. We further identify the emergent cognitive patterns that illustrate how models learn to *think with tools*. Finally, we report improved tool usage behavior with early code invocation and much more interactive turns with ASPO. Overall, our work provides the first principled explanation for TIR's success, shifting the focus from the mere fact *that* tools work to *why* and *how* they enable more powerful reasoning.

## 1 INTRODUCTION

Large language models (LLMs) have rapidly progressed from fluent generators to general-purpose problem solvers. Nevertheless, purely text-based reasoning often struggles with tasks that demand precise calculation, long-horizon search, faithful verification, or access to information beyond a model's parametric memory. As a powerful and empirically successful paradigm, Tool-Integrated Reasoning (TIR) (Feng et al., 2025; Li et al., 2025b) has emerged to address these limitations. Systems equipped with external tools have consistently and significantly outperformed their pure-text counterparts (OpenAI, 2025a;b; xAI, 2025). However, despite the widespread recognition of TIR's effectiveness, a principled account of the fundamental mechanisms, specifically *why* and *when* it helps, is still missing. Existing research has largely focused on demonstrating empirical success, leaving a crucial gap for a formal framework that can elucidate the origins of its benefits and define its capability boundaries.

To build such a framework, we first turn to reinforcement learning (RL) (Lambert et al., 2024; Sutton et al., 1998), the predominant paradigm for enhancing LLM reasoning. Recent theoretical work has established a critical consensus: in a pure-text environment, RL is constrained by an "invisible leash" (Wu et al., 2025). The learning process is largely confined to re-weighting probabilities within the base model's pre-existing trajectories, meaning it cannot discover fundamentally new reasoning trajectories that lie outside this initial capability (Yue et al., 2025).

The central thesis of this work is that tool integration fundamentally breaks this barrier. By introducing deterministic, non-linguistic state transitions via an external tool like a Python interpreter, TIR fundamentally expands the model's exploratory space. We provide the first formal proof that TIR enables a *strict expansion* of the model's empirical support, allowing it to generate correct trajectories that have negligible or even zero probability in a pure-text paradigm. Beyond theoretical reachability, we introduce the concept of *token efficiency* to argue that tools are a practical necessity. For any finite

token budget, there exist algorithmic strategies whose programmatic representations are concise, while their natural-language simulations are intractably verbose. Consequently, TIR unlocks a vastly larger *feasible support* of problem-solving strategies that are simply out of reach for pure-text models under realistic constraints. Extensions to other tools with informal propositions can be found in Appendix E.

We validate these theoretical claims through a series of experiments focusing on solving mathematical competition problems with a Python code interpreter. Our pass@$k$ analysis provides clear evidence that TIR decisively breaks the capability ceiling of pure-text models. Further investigation, using our proposed "algorithmic friendliness" metric, reveals that TIR's benefits are not confined to computationally-intensive problems but extend to those requiring significant abstract insight. Case studies of the model's behavior further illuminate *how* it leverages this expanded capability, revealing three emergent cognitive patterns: insight-to-computation transformation, exploration & verification via code, and offloading of complex calculations.

Finally, in exploring how to further optimize TIR models, we identify a practical algorithmic challenge: guiding model behavior, such as encouraging earlier tool use, via traditional reward shaping often leads to training instability in GRPO-like algorithms (Shao et al., 2024; Feng et al., 2025). To address this, we propose Advantage Shaping Policy Optimization (**ASPO**), a novel algorithm that circumvents the reward function and instead applies a stable, controllable bias directly to the advantage function. Our experiments show that ASPO successfully guides model behavior with early tool invocation and increased tool usages without compromising task performance or training stability.

Our contributions are as follows:

1. We provide the first formal theory for why TIR expands an LLM's capabilities, proving that it enables a strict expansion of both the feasible and empirical support compared to pure-text models.

2. We propose Advantage Shaping Policy Optimization (**ASPO**), a novel and stable algorithm for guiding the behavior of TIR models by directly shaping the advantage function, overcoming the instability of traditional reward-based methods.

3. We conduct a comprehensive empirical analysis that not only validates our theoretical claims and algorithm but also provides a mechanistic explanation of TIR's effectiveness, identifying its universal benefits across problem types and the emergent cognitive patterns it fosters.

## 2 RELATED WORK

A significant body of work focuses on developing RL frameworks for strategic tool use. Feng et al. (2025) propose ReTool, an RL-based framework that demonstrates high data efficiency for learning tool use. Similarly, Li et al. (2025b) introduce ToRL, a method designed to address the challenges of scaling tool-integrated RL to more complex and demanding scenarios. Bai et al. (2025) document methods for effective code-integrated reasoning. This paradigm shares the goal of augmenting LLM reasoning with external Python execution. Focusing on training from base models, Xue et al. (2025) present SimpleTIR, an end-to-end framework for multi-turn TIR that enables stable training from scratch, a process they refer to as the "Zero" setting.

Beyond these RL-for-tool frameworks, other research investigates the limitations of RL for pure-text reasoning. Yue et al. (2025) empirically find that RL does not incentivize novel reasoning capacity. Providing a theoretical framework to explain such findings, Wu et al. (2025) propose the "invisible leash" theory, suggesting that models may struggle to discover reasoning paths outside their original knowledge distribution.

Beyond programmatic tools like Python interpreters, another line of work integrates search engines to equip LLMs with up-to-date knowledge via RL. Jin et al. (2025) propose Search-R1, where LLMs interleave reasoning with real-time queries, trained with outcome-based rewards and stabilized by masking retrieved tokens, achieving strong multi-turn QA performance. To tackle uncertainty in complex web tasks, Li et al. (2025a) introduce WebSailor, a post-training method that narrows the gap with proprietary agents. In this work, we primarily focus on utilizing Python interpreters to enhance the LLM's ability to solve complex reasoning problems in mathematics; similar principles apply for enhancing knowledge-seeking ability and we have informal discussions in Appendix E.

## 3 METHOD

In this section, we formalize the argument that integrating an external computational tool, such as a code interpreter, fundamentally enhances a Large Language Model's (LLM) capabilities. We structure our argument in two parts. First, we provide a formal proof demonstrating that tool integration results in a strict expansion of the model's generative support, thereby breaking the "invisible leash" that constrains purely text-based models (Wu et al., 2025). Second, we introduce the concept of *token efficiency* to argue that even for problems theoretically solvable by text-based models, tool integration is a practical necessity for expressing complex algorithms within any feasible token budget. It is worth emphasizing that our aim here is not to obtain surprising complexity-theoretic separations, but to introduce a precise vocabulary and analytical lens missing in prior work, which enables systematic reasoning about tool-augmented models.

### 3.1 FORMAL PROOF: SUPPORT EXPANSION VIA TOOL INTEGRATION

We begin by establishing that augmenting an LLM with a deterministic external tool strictly expands its support, enabling it to generate trajectories that were previously impossible.

#### 3.1.1 THEORETICAL CONTEXT: THE LIMITS OF STANDARD RL

To ground our proof, we first adopt the theoretical framework proposed by Wu et al. (2025), which formalizes the limitations of standard on-policy reinforcement learning (DeepSeek, 2025; Lambert et al., 2024; Schulman et al., 2017) on training LLMs. We briefly introduce the key concepts (a detailed review is provided in Appendix B).

The **support** of a model with distribution $p$, $\text{supp}(p)$, is the set of all trajectories it can generate with non-zero probability. The central limitation of RLVR, as established by Wu et al. (2025), is the **Support Preservation Theorem**. It states that the support of the RL-trained policy is a subset of the support of the base model. The Support Preservation Theorem formalizes the "invisible leash": RLVR can only re-weight probabilities within the model's pre-existing support, but cannot expand it.

A more practical variant of support $\text{supp}(p)$ is the **empirical support**, $\text{supp}_\varepsilon(p)$, which only includes trajectories with a probability greater than a small threshold $\varepsilon$; in what follows, all statements will be made under this empirical-support view, within which we establish a strictly stronger and practical result on TIR models.

#### 3.1.2 PROOF OF SUPPORT EXPANSION

We consider two types of LLMs in this work. A pure-text model is a standard language model with distribution $q_{\text{text}}$. We compare this to a TIR model with distribution $p_{\text{TIR}}$, which pairs a language model with a deterministic external oracle (e.g., a Python interpreter). Our goal here is to study the reachable trajectory space induced by the external tool, not the outcome of RL training on tool use. Therefore, throughout this section we assume that the pure-text model $q_{\text{text}}$ and the TIR model $p_{\text{TIR}}$ share exactly the same underlying language model parameters. The only difference is that the TIR model is equipped with additional deterministic transitions provided by the tool. Now we present the main theorem and its proof sketch (a complete proof is provided in Appendix C):

> **Theorem 3.1** (Strict Expansion of Empirical Support via Tool Integration). *There exists an $\varepsilon > 0$ and a family of problem instances such that the empirical support of a pure-text model is a strict subset of the empirical support of a tool-integrated model:*
>
> $$\text{supp}_\varepsilon(q_{\text{text}}) \subset \text{supp}_\varepsilon(p_{\text{TIR}}).$$

*Proof Sketch.* (**Inclusion** $\subseteq$) is straightforward. Because, given that both models share the same underlying language-model parameters, the TIR model can reproduce any natural-language trajectory generated by the pure-text model with exactly the same probability. Thus every pure-text trajectory remains reachable in the TIR environment with unchanged probability. (**Strictness** $\subset$) relies on a constructive proof using a standard cryptographic primitive: *random oracle*. The tool-integrated model can deterministically solve the oracle problem in a single step. In contrast, the pure-text

model must guess the high-entropy $m$-bit output, succeeding with a probability $(2^{-m})$ that becomes negligible for any practical threshold $\varepsilon$. Thus, a correct trajectory exists that is within the empirical support of $p_{\text{TIR}}$ but not of $q_{\text{text}}$. □

The use of a random oracle here is purely idealized and serves as an existence proof: it witnesses that there exist deterministic external tools for which no pure-text next-token model can assign non-negligible probability to the correct trajectory. We further show in Appendix C how the same argument extends to realistic tools.

We have shown that $\text{supp}(q_{\text{text}})$ is a strict subset of $\text{supp}(p_{\text{TIR}})$. Unlike pure-text models, which are constrained by Support Preservation Theorem, tool integration breaks the "invisible leash" by introducing new, deterministic state transitions, thereby creating a strict expansion of the model's support.

### 3.2 TOKEN EFFICIENCY AND FEASIBLE SUPPORT UNDER A BUDGET

The proof in the previous section establishes that a tool-integrated model can generate trajectories that are impossible for a pure-text model. This, however, raises a deeper question: can a pure-text model achieve the same outcomes by *simulating* the computational process through natural language? While the resulting trajectories $y$ may differ syntactically ($y_{\text{text}} \neq y_{\text{TIR}}$), they might represent the *same* underlying problem-solving strategy. To properly evaluate this, we must move beyond comparing trajectories based on string identity and instead assess them on their semantic content and efficiency. This motivates our analysis of *token efficiency*.

#### 3.2.1 THE CONCEPT OF TOKEN EFFICIENCY

A key distinction between programmatic and natural language solutions is their *token efficiency*: the compactness with which a solution is represented. For any task involving iteration or recursion, a programmatic representation offers a scalable, abstract description with a near-constant token cost, e.g., $O(1)$. In contrast, a natural language trace that simulates the same process must enumerate each computational step, leading to a token cost that scales with the size of the computation. The tables in Appendix F illustrate this stark disparity for common algorithmic patterns: simple iteration (Table 1), large linear systems (Table 2), dynamic programming (Table 3), and graph search (Table 4). In each case, the programmatic solution remains a concise, scalable representation, while the natural language simulation becomes a verbose, concrete enumeration that is untenable for non-trivial problem sizes.

#### 3.2.2 FEASIBLE SUPPORT UNDER A TOKEN BUDGET

The fundamental disparity in token efficiency motivates a more *practical*, *budget-aware* analysis of a model's capabilities, moving beyond theoretical possibilities to what is achievable within operational constraints. To formalize this, we first define the total *token cost* of a trajectory, $\text{cost}(y)$, as the sum of all tokens consumed (i.e., prompt, model generation, and tools I/O), which must not exceed the model's context budget $B$. This allows us to define the set of strategies a model can feasibly execute:

**Definition 3.2** (Computational Equivalence Class). Two trajectories, $y_1$ and $y_2$, are computationally equivalent, denoted $y_1 \sim y_2$, if they solve the same problem $x$ by implementing the same core algorithm. This relation partitions the space of all trajectories $\mathcal{Y}$ into equivalence classes, where each class $[y]$ represents a distinct algorithmic "idea" or "strategy".

**Definition 3.3** (Feasible Support under Budget $B$). An algorithmic strategy, represented by equivalence class $[y]$, is within the feasible support of a model $M$ under token budget $B$, denoted $[y] \in \text{supp}_B(M)$, if and only if there exists at least one trajectory $y' \in [y]$ such that $M(y'|x) > 0$ and its token $\text{cost}(y')$ does not exceed the budget:

$$\exists y' \in [y] \text{ s.t. } M(y'|x) > 0 \text{ and } \text{cost}(y') \leq B.$$

This definition captures a model's practical ability to realize a problem-solving strategy within operational constraints. With this formal framework in place, we can now state our central claim regarding the practical supremacy of tool-integrated models and its proof sketch (a detailed proof is provided in Appendix D):

**Theorem 3.4** (Strict Supremacy of Tool-Integrated Feasible Support). *For any non-trivial algorithmic problem class and any token budget $B$, there exists a problem size $n_B$ such that the feasible support of a pure-text model is a strict subset of the feasible support of a tool-integrated model:*

$$supp_B(q_{text}) \subset supp_B(p_{TIR}).$$

*Proof Sketch.* **(Inclusion $\subseteq$)** holds because a tool-integrated model can always operate within the pure-text paradigm. **(Strictness $\subset$)** follows directly from the divergent scaling properties of the natural language. For any finite budget $B$, we can choose a problem size $n_B$ large enough that the token cost of a natural language simulation exceeds $B$, while the $O(1)$ programmatic representation remains well within budget. Thus, the algorithmic strategy is feasible for $p_{\text{TIR}}$ but not for $q_{\text{text}}$. $\qquad\square$

The theorem crystallizes the practical implications of token efficiency. It establishes that for any finite computational budget, there is a vast class of algorithmic strategies that pure-text models are fundamentally incapable of executing. Not because the solution is unknowable, but because its expression in natural language is too *verbose*. Tool integration is therefore not merely a convenience; it is a necessity for expanding the set of algorithmic approaches that LLMs can feasibly deploy. This provides a strong argument for a paradigm where LLMs act as reasoning engines that delegate complex computational tasks to specialized, efficient tools.

The analysis above focuses on token efficiency because the context window is the primary hard constraint for LLMs: a strategy is only usable if it can be represented within the available tokens, regardless of its external computation time. Although external tools do incur computational cost, in practical systems they are introduced exactly because they execute specific tasks much more efficiently than simulating the same process through natural-language traces. It ensures that TIR remains more efficient when accounting for the computational cost of the tool. As a result, incorporating tool cost does not change the underlying conclusion of this section.

### 3.3 Algorithmic Improvement: Advantage Shaping for Early Code Invocation

The TIR models often default to a conservative strategy: completing the majority of their abstract reasoning via text before invoking the code interpreter for the final-step calculation or verification. This overlooks a potentially more powerful paradigm where the interpreter is used as an exploratory tool throughout the reasoning process. We hypothesize that encouraging the model to invoke code *earlier* could foster a more dynamic, flexible, and hypothesis-driven reasoning style, potentially unlocking novel problem-solving strategies.

To encourage earlier tool use, we first tried adding an early-code bonus to the reward function, but this approach proved highly unstable. In GRPO-like algorithms, group normalization cancels the primary correctness signal when all samples in a group are correct, catastrophically amplifying the auxiliary bonus and distorts the learning objective (see Appendix G for a detailed analysis).

To circumvent the distorting effects of reward normalization, we developed a more robust method that we term Advantage Shaping Policy Optimization (**ASPO**). Instead of manipulating the reward, we directly modify the final advantage value after the standard correctness-based advantage $A_{\text{correct}}$ has been calculated. For any response $i$ that is both correct and contains code, we compute the new advantage $A_i$ as follows:

$$A_i = A_{\text{correct},i} + \text{clip}\left(\delta \cdot \frac{p_i - \text{mean}(\mathbf{p})}{\text{mean}(\mathbf{L})}, \ -k \cdot A_{\text{correct},i}, \ k \cdot A_{\text{correct},i}\right),$$

where $\mathbf{p}$ and $\mathbf{L}$ are the sets of first code invocation positions and total response lengths for all correct, code-containing responses within the group. Furthermore, $\delta$ is a negative coefficient to encourage early code invocation, and $k$ is a clipping hyperparameter that bounds the magnitude of auxiliary advantage within a proportion of the basic advantage of correctness. Importantly, the auxiliary term is applied only to trajectories that are both correct and contain tool call; trajectories with incorrect final answers never receive any bonus, regardless of their tool-use behavior.

This formulation has several key merits, primarily by circumventing the uncontrollable effects of advantage normalization inherent to reward-based modifications. First, it addresses a critical flaw in the reward-based approach: the inability to guarantee a positive advantage for all correct answers.

After adding the auxiliary reward, a correct response's total reward could fall below the group average, leading to a negative GRPO normalized advantage, which effectively punishes a correct solution. Second, the GRPO normalization process introduces uncontrollable volatility: the $\text{std}(\mathbf{R})$ in the denominator unpredictably scales the auxiliary signal, making its influence inconsistent across groups.

Our ASPO algorithm resolves both issues. By applying a clipped bias directly to $A_{\text{correct}}$, we ensure the final advantage remains positive and that the early-code incentive is always a subordinate nudge, never overwhelming the primary objective of correctness. Furthermore, this approach bypasses the volatile scaling effect of $\text{std}(\mathbf{R})$ entirely. Besides, the auxiliary advantage involves $p - \text{mean}(p)$. As a result, within the correct-and-tool subgroup, ASPO essentially performs a zero-sum redistribution of advantage (ignoring clipping effects). It does not artificially inflate the total advantage mass for this subgroup. The total advantage mass of "tool-use" or "correct-answer" also remains unchanged. From this perspective, it does not directly encourage more tool calls. Rather, it re-ranks the responses in the correct-and-tool subgroup, expressing a preference for early invocation over late invocation among the correct-and-tool responses. Finally, the choice to normalize the code invocation position by the mean response length $\text{mean}(\mathbf{L})$ rather than the standard deviation of positions $\text{std}(\mathbf{p})$ is deliberate. The latter is unstable: when invocation positions in a group are tightly clustered, a small $\text{std}(\mathbf{p})$ would excessively amplify the signal, whereas a more stable denominator like $\text{mean}(\mathbf{L})$ is consistent and meaningful. This method allows us to stably and effectively encourage early code invocation, the empirical results of which are detailed in Section 4.4.

In essence, ASPO provides a general and robust framework for guiding a model's behavior towards desired styles or properties without compromising the primary learning objective (e.g., accuracy). By directly manipulating the advantage values, ASPO avoids the instabilities that can arise from altering the reward function, particularly in GRPO-like algorithms that rely on reward normalization. This method ensures that the incentive for the desired behavior (in this case, earlier code invocation) acts as a stable adjustment. The core principles of ASPO could be *readily adapted* to encourage other desirable behaviors in a variety of scenarios, offering a reliable approach to shape model conduct while preserving training stability and overall task performance.

## 4 EXPERIMENTS

All experiments are based on the Qwen3-8B model (Qwen, 2025). We compare our proposed Tool-Integrated Reasoning (TIR) model against a pure-text RL baseline (see Figure 5 in Appendix H). For training, we used a 10,000-problem subset of the DAPO dataset (Yu et al., 2025), which is sufficient for our goal of understanding TIR mechanisms rather than achieving state-of-the-art benchmark scores. Both models were trained using the DAPO algorithm (Yu et al., 2025), a variant of GRPO (DeepSeek, 2025). Our primary evaluation benchmarks are AIME24, AIME25, and Omni-MATH-512, a curated set of 512 challenging problems from the Omni-MATH dataset (Gao et al., 2024). A detailed experimental setup is provided in Appendix H.

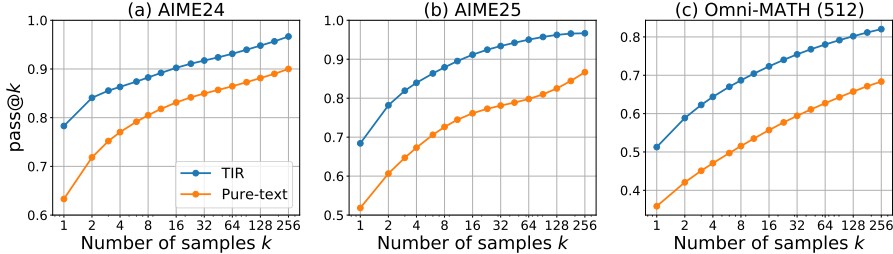

Figure 1: Pass@$k$ curves for the TIR (RL trained) and pure-text models (Qwen3-8B) across three benchmarks: (a) AIME24, (b) AIME25, and (c) Omni-MATH-512. The pure-text model here corresponds to the vanilla Qwen3-8B base model without RL fine-tuning. This choice is intentional and conservative: prior work (Yue et al., 2025) reports that RL often decreases pass@$k$ performance at large $k$. We also evaluated the AIME25 pass@$k$ for RL-trained pure-text model, finding that the vanilla Qwen3-8B model performs better than its RL counterpart at large $k$. We therefore adopt the stronger vanilla baseline to better characterize the capability ceiling. The detailed numerical data corresponding to this figure are provided in the Appendix I.

## 4.1 PASS@$K$ EXPERIMENTS: TIR BREAKS THE CAPABILITY CEILING

To empirically test our theoretical claims, this section investigates whether TIR can overcome the capability ceiling observed in pure-text models (Yue et al., 2025; Wu et al., 2025). Similar to Yue et al. (2025) and others, we use the pass@$k$ metric, with low-variance estimation from Chen et al. (2021), as it provides a robust measure of a model's underlying problem-solving potential.

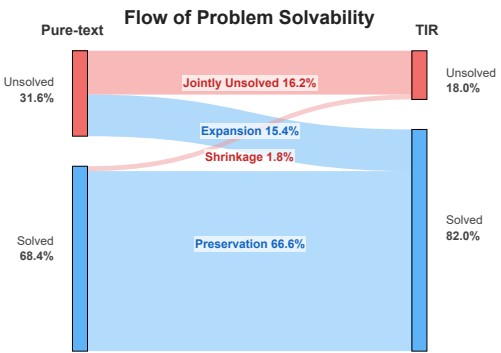

Figure 1 presents the macroscopic evidence from our experiments. It plots the pass@$k$ curves for both the TIR model (RL trained) and the pure-text baseline (Qwen3-8B) across our three evaluation benchmarks, with the max $k$ of 256. The results are unequivocal: on AIME24, AIME25, and Omni-MATH-512, the TIR model's performance curve is consistently and significantly higher than that of the pure-text model. Crucially, we observe *no intersection* between the curves, even as $k$ increases to 256. This stands in stark contrast to previous findings where RL-trained text models, while improving pass@1, often do so at the cost of the broader capability envelope, eventually being surpassed by the base model at high values of $k$ (Yue et al., 2025). Our results show TIR does not suffer from this trade-off; it elevates the *entire* pass@$k$ curve. We also show that this phenomenon is stable under different temperature in Appendix J

Figure 2: The flow of problem solvability on Omni-MATH-512 when transitioning from the pure-text model to the TIR model, evaluated at $k = 256$. A detailed version is provided in Appendix K.

To further understand this performance gain at a per-problem level, we visualize the "flow of solvability" on the Omni-MATH-512 dataset in Figure 2. This Sankey diagram illustrates how the solvability status of individual problems changes when moving from the pure-text model to the TIR model (samples $k = 256$ responses per problem). We categorize the problems into four distinct groups as Wu et al. (2025): **Capability Expansion**: Problems the pure-text model fails to solve but the TIR model succeeds on; **Capability Preservation**: Problems solved by both models; **Capability Shrinkage**: Problems solved by the pure-text model but not by the TIR model; **Jointly Unsolved**: Problems that neither model can solve. The diagram reveals a massive net gain in problem-solving capability. The Capability Expansion set contains 15.4% problems, whereas the Capability Shrinkage set contains only 1.8% (In Appendix L we show that this small shrinkage is due to sampling variance and RL training). This provides direct empirical validation for our theoretical argument in Section 3, demonstrating that TIR facilitates a practically significant expansion of the model's effective support.

In summary, both macroscopic pass@$k$ analysis and microscopic problem-level tracking confirm that tool-integrated reasoning decisively breaks the capability ceiling of its pure-text counterpart, enabling the model to solve a wide range of problems that were previously out of its reach.

## 4.2 BENEFITS OF TIR EXTEND BEYOND COMPUTATIONALLY-INTENSIVE PROBLEMS

A crucial question arises from our initial findings: is the observed capability expansion of TIR merely an *artifact* of solving problems that are inherently algorithmic? The most direct yet naive interpretation of TIR's success is that it simply offloads complex arithmetic, acting as a superior *calculator*. However, a more nuanced counterargument posits that TIR's effectiveness, while beyond simple calculation, is still confined to problems whose structure can be directly mapped to a known algorithm such as exhaustive search in combinatorics. This perspective suggests that TIR improves the model's capability on problems that are computationally-intensive or inherently algorithmic, but offers *little advantage* when the problem is highly abstract.

To rigorously test our hypothesis, we first introduce the concept of "algorithmic friendliness", which is defined as a measure of how reliant a problem's solution is on standard computation *versus* deep mathematical insight. To operationalize this concept, we developed a detailed five-point rubric for classifying problems, as presented in Appendix M. This scale ranges from a score of 1 for problems

that are fundamentally abstract and non-computational, to 5 for those solvable by a direct application of a textbook algorithm. We then applied this rubric to classify each problem in the Omni-MATH-512 dataset. This classification was performed by providing both the problem statement and its solution idea to the Gemini 2.5 Pro API (Gemini, 2025), which then assigned a score based on the rubric. The resulting distribution of problem types, shown in Figure 8(f) (Appendix N), reveals a crucial characteristic of the dataset. Contrary to being biased towards highly algorithmic problems, the distribution's peak is concentrated in the medium friendliness categories (scores 2, 3 and 4). This confirms that Omni-MATH-512 serves as a fair and challenging testbed for our analysis, not one skewed towards problems with simple computational solutions. To check this classification, we manually inspected all problems in the lowest-friendliness bucket (24 items with scores between 1.0 and 1.5) and found them consistent with the rubric. We also queried Gemini multiple times per problem and averaged the scores to reduce stochastic variation. Since the friendliness lables are only used for grouping, rather than as supervision or evaluation signals, this level of validation is sufficient for our analysis.

Figure 8(a)-(e) (Appendix N) presents our *core findings*. It displays the pass@$k$ curves for the TIR and pure-text models, grouped by the algo friendliness of the problems. As expected, the performance gap between the two models is most pronounced for problems with high friendliness (scores 4.0-5.0), where TIR's ability to execute algorithms directly provides a massive advantage (Figure 8(d),(e)). The *most critical* finding, however, comes from the lowest friendliness group (scores 1.0-2.5). Even for these problems, which depend heavily on abstract reasoning and are ill-suited to direct computation, the TIR model maintains a significant and consistent performance advantage over the pure-text baseline, outperforming it by approximately 9% in pass@256 accuracy (Figure 8(a),(b)).

This result demonstrates that the benefits of TIR are not confined to easily programmable problems. The tool serves a more profound purpose than acting as a simple calculator or a direct algorithm-implementer. It suggests that the model is leveraging the code interpreter in more complex and sophisticated ways, which we will investigate in the next subsection.

### 4.3 EMERGENT COGNITIVE PATTERNS OF TOOL INTEGRATION

To understand *how* TIR is effective beyond purely algorithmic problems, our qualitative analysis identified three recurring patterns of code utilization. These patterns reveal a sophisticated interplay where the model is not just *using* a tool, but fundamentally *thinking with* it. **Pattern 1: Insight-to-computation transformation.** The model first engages in text-based reasoning to transform an abstract problem into a computationally tractable form. It then invokes the interpreter to execute a genuine algorithm (e.g., search, enumeration, DP) on this newly formulated sub-problem. **Pattern 2: Exploration and verification via code.** For problems with unclear solution paths, the model uses the interpreter as an interactive sandbox. It formulates conjectures, writes short code snippets to test them, and iteratively refines its strategy based on the feedback, allowing it to discover insights through empirical experimentation. **Pattern 3: Offloading complex calculation.** In the most direct usage, the model delegates complex or tedious calculations to the interpreter. This minimizes the risk of unforced computational errors that could derail a correct line of reasoning. The first two patterns represent a fundamental departure from pure-text reasoning, constituting new *Computational Equivalence Classes* that are infeasible for pure-text models due to prohibitive token costs (i.e., they lie outside the *Feasible Support under Budget B*). Such dynamic and flexible code invocation enables the TIR model to break the capability ceiling of its pure-text counterpart. Detailed analysis and examples of these patterns are provided in Appendix O.

### 4.4 EMPIRICAL ANALYSIS OF ASPO FOR EARLY CODE INVOCATION

In this section, we empirically validate our ASPO algorithm, designed to encourage earlier code invocation. We aim to answer two primary questions: (1) Does this method maintain training stability and final task performance, unlike the naive reward-based approach? (2) Does it effectively and controllably alter the model's tool-use behavior as intended? We test our baseline model against the unstable reward-based approach and two variants of our ASPO algorithm: a *conservative* setting ($\delta = -2.0, k = 0.7$) and an *aggressive* setting ($\delta = -2.5, k = 0.9$).

**Stability and performance remain uncompromised.** Figure 3 provides a clear validation of our method's stability. As mentioned in our analysis in Section 3, the naive reward-based approach

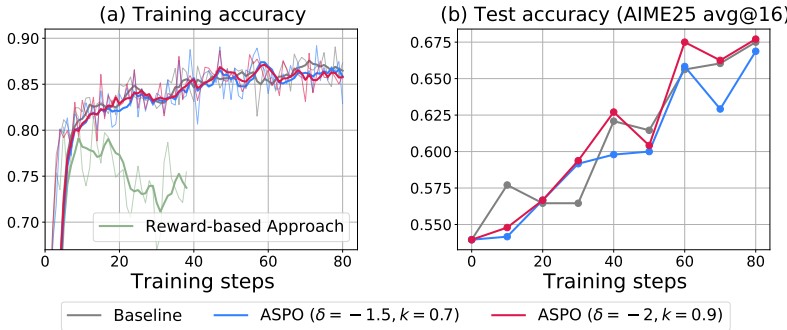

Figure 3: The (a) training and (b) testing accuracy of the baseline (the standard DAPO algorithm (Yu et al., 2025) with only final-answer reward) and ASPO algorithm.

quickly becomes unstable, causing the training reward to collapse (Figure 3(a)). In stark contrast, the training curves for our ASPO algorithm with both conservative and aggressive settings remain stable and almost perfectly aligned with the baseline. Furthermore, this stability does not come at the cost of final performance. Figure 3(b) shows that the final "avg@16" accuracy on AIME25 for both variants is statistically indistinguishable from the baseline. This is a crucial result: our method successfully avoids the pitfalls of reward modification, ensuring that the primary goal of solving the problem correctly is not sacrificed.

**A significant shift in cognitive behavior.** Having established the method's safety, we now demonstrate its effectiveness in reshaping the model's reasoning strategy. Figure 4 presents a comprehensive analysis of the model's code-use behavior on AIME25, averaged over 16 responses per question. The results show a dramatic and targeted shift. The most significant change is in the code invocation timing (Figure 4(b)), where the average position of the first code call is brought forward from 4,000 tokens in the baseline down to 1,000 tokens. Concurrently, the model becomes a much more active tool user: the average number of code rounds per problem more than doubles, from 1.3 to 3.3 (Figure 4(e)), and the code ratio approaches nearly 100%, indicating that using the interpreter becomes a default part of the model's process (Figure 4(c)). This behavioral shift is starkly evident when examining the distribution of responses for a single challenging problem. For instance, on Q30 of the AIME25, the baseline model exhibited reluctant and inconsistent tool use: out of 16 independent responses, four failed to make a single code call, and the median number of invocations was just 2. In stark contrast, our ASPO-trained model integrated the tool as an indispensable part of its problem-solving process. It invoked the code in all 16 responses for the same problem, and the median number of tool calls increased from 2 to 13. More significantly, a quarter of the responses demonstrated highly iterative behavior, making more than 20 tool calls, which is entirely absent in the GRPO-trained baseline. This shows a clear transformation from a conservative, late-stage "calculator" usage pattern to an early, iterative, and exploratory "interactive partner" paradigm.

**Controllability and absence of reward hacking.** Importantly, this behavioral shift is achieved without inducing reward hacking. We manually inspected a large number of samples and found no instances of the model inserting trivial or meaningless code early in its response merely to satisfy the incentive. The stability of the final task accuracy (Figure 3(b)) and the code pass ratio (Figure 4(d)) further substantiates this. Finally, the difference between the conservative and aggressive settings demonstrates that the degree of behavioral change is tunable via the hyperparameters $\delta$ and $k$.

## 5 CONCLUSIONS

In this work, we presented a comprehensive investigation into the foundational mechanisms of Tool-Integrated Reasoning (TIR). We moved beyond empirical demonstrations to establish a formal theoretical framework explaining its effectiveness. Our core theoretical contribution is the proof that TIR enables a strict expansion of both the empirical and feasible support of an LLM, breaking the "invisible leash" that constrains pure-text models and making complex algorithmic strategies practically achievable within finite token budgets. On the algorithmic front, we identified the instability of reward shaping for guiding model behavior in TIR systems and proposed Advantage

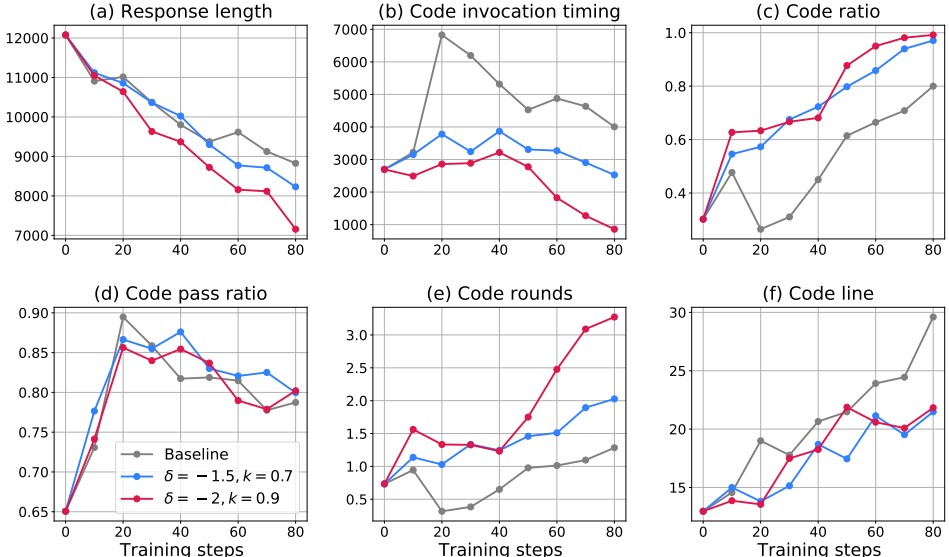

Figure 4: Evaluation results of the baseline and ASPO algorithm on AIME25. (a) Response length, (b) code invocation timing, (c) code ratio, (d) code pass ratio, (e) code rounds and (f) code lines.

Shaping Policy Optimization (ASPO), a stable and effective alternative that directly modifies the advantage function.

Our experiments provided strong empirical validation for these claims. We demonstrated that TIR model equipped with a Python interpreter decisively surpasses the performance of pure-text models across challenging mathematical reasoning benchmarks. Our analysis, using a novel "algorithmic friendliness" metric, revealed that TIR's benefits are universal, extending even to problems that are highly abstract and less amenable to direct computation. Qualitative analysis further uncovered the sophisticated, emergent cognitive patterns that arise from the synergy between LLM reasoning and tool execution.

Ultimately, our findings advocate for a paradigm shift: viewing LLMs not as monolithic problem-solvers, but as core reasoning engines that intelligently delegate computational tasks to specialized, efficient tools. The principles and methods developed here, particularly ASPO, open avenues for more nuanced and stable control over the behavior of powerful tool-integrated agents.

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

## A  LLM USAGE

During the preparation of this work, we used LLM to aid in polishing the manuscript and improving language. All final content was reviewed and revised by the authors to ensure its accuracy and originality. The core ideas, methods, and conclusions of the paper are solely the work of the authors.

## B   THE THEORETICAL BACKGROUND

To ground our proof, we adopt the theoretical framework proposed by Wu et al. (2025), which formalizes the limitations of standard on-policy reinforcement learning (DeepSeek, 2025; Lambert et al., 2024; Schulman et al., 2017) on training LLMs.

**Definition B.1** (Support of a Model (adapted from (Wu et al., 2025))). Let $\mathcal{Y}$ be the space of all possible generative trajectories. The support of a model with distribution $p(y|x)$ is the set of all trajectories that can be generated with a non-zero probability for a given prompt $x$:

$$\text{supp}(p) := \{y \in \mathcal{Y} \mid p(y|x) > 0\}$$

**Definition B.2** (Empirical Support (from Wu et al. (2025))). For a threshold $\varepsilon > 0$, define the empirical support of $p$ as

$$\text{supp}_\varepsilon(p) := \{y \in \mathcal{Y} \mid p(y|x) \geq \varepsilon\}.$$

This definition is central to understanding a model's intrinsic capabilities. The following theorem from Wu et al. (2025) establishes a key constraint for models trained with Reinforcement Learning from Verifiable Rewards (RLVR) (Lambert et al., 2024; DeepSeek, 2025):

> **Theorem B.3** (Support Preservation under RLVR (from Wu et al. (2025))). *Let $\pi_\theta(y|x)$ be an RLVR-trained policy distribution initialized from a base model with distribution $q(y|x)$. For any prompt $x$, the support of the trained policy is a subset of the support of the base model:*
>
> $$supp(\pi_\theta) \subseteq supp(q)$$
>
> *This implies that if $q(y^*|x) = 0$ for a correct trajectory $y^*$, then RLVR can never discover $y^*$.*

Theorem B.3 formalizes the "invisible leash": RLVR can only re-weight probabilities within the model's pre-existing support. We next show a strictly stronger, practical statement under an empirical-support view.

## C   THE DETAILED PROOF OF SUPPORT EXPANSION

We consider two types of LLMs in this work. A pure-text model is a standard language model with distribution $q_{\text{text}}$ that generates tokens exclusively from its vocabulary $\mathcal{V}$. We compare this to a tool-integrated model, a system $(M, \mathcal{O})$ with distribution $p_{\text{TIR}}$, which pairs a language model $M$ with a deterministic external oracle $\mathcal{O}$ (e.g., a Python interpreter). The generative process for this model includes not only probabilistic token generation from $\mathcal{V}$ but also deterministic tool-use transitions. In such a transition, the model $M$ emits a tool call $y_{\text{call}}$, the oracle executes it, and the resulting output $y_{\text{out}} = \mathcal{O}(y_{\text{call}})$ is deterministically returned as the next state. Our goal here is to study the reachable trajectory space induced by the external tool, not the outcome of RL training on tool use. Therefore, we assume that the pure-text model and the TIR model share exactly the same underlying language model parameters. The only difference is that the TIR model is equipped with additional deterministic transitions provided by the tool.

Now we present the detailed proof of Theorem 3.1.

*Proof.* The proof proceeds in two parts. First, we establish the subset relationship ($\subseteq$), and second, we prove the relationship is strict ($\neq$) by demonstrating the existence of trajectories accessible only to the tool-integrated model.

**Part 1: Proving supp($q_{\text{text}}$) $\subseteq$ supp($p_{\text{TIR}}$)**

Let $y$ be an arbitrary trajectory in the support of the pure-text model, such that $q_{\text{text}}(y|x) > 0$. The trajectory $y$ consists exclusively of tokens from the vocabulary $\mathcal{V}$. Because, given that both models share the same underlying language-model parameters, the TIR model can reproduce any natural-language trajectory generated by the pure-text model with exactly the same probability. Thus every pure-text trajectory remains reachable in the TIR environment with unchanged probability. Therefore, for any $y \in \text{supp}(q_{\text{text}})$, it follows that $y \in \text{supp}(p_{\text{TIR}})$, establishing that $\text{supp}(q_{\text{text}}) \subseteq \text{supp}(p_{\text{TIR}})$.

**Part 2: Proving Strictness**

To prove strictness, we use a constructive approach based on a standard cryptographic primitive: a *random oracle*. Let us consider a problem instance where the solution requires computing $y_{out} = H(x)$, where $H$ is a random oracle. A random oracle is a theoretical black box that, for any new input query, returns an output chosen uniformly at random from its output space (e.g., $\{0, 1\}^m$), but deterministically returns the same output for repeated queries of the same input. This construction is theoretically convenient and serves as an idealization of practical cryptographic hash functions (e.g., SHA-256). For a model without access to the oracle, its only strategy to find $y_{out}$ is to guess it. The probability of correctly guessing a specific $m$-bit string is $2^{-m}$.

Now, consider a trajectory $y^* = (y^*_{prefix}, y_{out}, y^*_{suffix})$ that involves computing $H(x)$. We assume the underlying language model for both $p_{TIR}$ and $q_{text}$ is identical. The tool-integrated model $p_{TIR}$ can invoke the oracle to obtain $y_{out}$ deterministically. In contrast, the pure-text model, $q_{text}$, must guess $y_{out}$ from an output space of size $2^m$, succeeding with a probability of only $2^{-m}$. Thus, the total probabilities of producing $y^*$ are directly related:

$$q_{text}(y^*|x) = p_{TIR}(y^*|x) \cdot 2^{-m}.$$

For any non-negligible probability $p_{TIR}(y^*|x)$ and a sufficiently large $m$, the corresponding $q_{text}(y^*|x)$ becomes arbitrarily small. We can therefore always choose an $\varepsilon$ such that $q_{text}(y^*|x) < \varepsilon \leq p_{TIR}(y^*|x)$. So we find that $y^* \notin \text{supp}_\varepsilon(q_{text})$ while $y^* \in \text{supp}_\varepsilon(p_{TIR})$. This establishes strictness. $\qquad\square$

### On the Use of Random Oracle

The use of a random oracle in the strictness proof above is intentional and purely idealized. Its purpose is existential: it demonstrates that there exist deterministic external transitions for which no pure-text model can assign non-negligible probability to the correct trajectory, while a TIR model retains a deterministic path.

While idealized, the random oracle proof can map directly to realistic scenarios and tools. We outline two representative examples.

1. Heavy numerical computation. The core idea is that the probability of pure-text model generating correct computation results decreases as computational tasks increase in complexity or length. Consider a family of problems whose solution requires executing a sequence of $N$ arithmetic or numerical operations (e.g., iterative numerical solvers, time-stepping PDEs). Let a pure-text model execute each primitive operation correctly with probability $p < 1$. Then the probability of emitting a correct computational result decays exponentially. A TIR model, equipped with a deterministic computational tool (e.g., numerical solver or Python interpreter), obtains the correct result via tool call whose success probability does not depend on $N$.

2. Knowledge Retrieval. Querying an external database for unknown facts (e.g., specific weather data) is mathematically equivalent to a Random Oracle. A pure-text model can only guess the correct entry, with success probability decreasing as the answer length increases. In contrast, a TIR model that issues a structured query to the database retrieves the correct entry deterministically.

These examples show that the random-oracle construction is not an unrealistic corner case, but a convenient abstraction capturing a general principle: There exist practically relevant families of problems whose solution probabilities for pure-text model vanish with problem size, while a TIR model retains a deterministic solution trajectory. Thus, the strict support-expansion theorem captures a structural phenomenon applicable well beyond idealized oracle settings.

## D    THE DETAILED PROOF OF FEASIBLE SUPPORT SUPREMACY

Here we present the detailed proof of Theorem 3.4.

*Proof.* The proof requires showing both inclusion ($\subseteq$) and strictness ($\neq$).

**Inclusion ($\subseteq$):** Any algorithmic strategy that is feasibly executable by a pure-text model within budget $B$ is, by definition, also executable by a tool-integrated model that simply abstains from using its tool.

**Strictness ($\neq$):** We must show there exists an algorithmic class $[y_A]$ in $\text{supp}_B(p_{TIR})$ but not in $\text{supp}_B(q_{text})$. This follows directly from the divergent scaling properties of natural language versus

programmatic representations, as illustrated in Tables 1-4. For any algorithm whose pure-text simulation cost scales with problem size $n$ (e.g., $\Omega(n)$, $\Omega(V + E)$), we can choose a size $n_B$ such that the cost exceeds any finite budget $B$. The programmatic representation, costing $O(1)$, remains within budget. Thus, for a sufficiently large problem size, the corresponding algorithmic classes are in the feasible support of $p_{\text{TIR}}$ but not $q_{\text{text}}$, proving strict inclusion. □

# E    EXTENSIONS TO OTHER TOOLS AND INTERACTIONS WITH ENVIRONMENTS

Our arguments in Sections 3.1.2 and 3.2 extend beyond Python to a broad family of external tools and interactive settings. At a high level, any interface that (i) affords *state transitions* not expressible by next-token sampling alone and/or (ii) delivers *high information per token of I/O* will both expand support (Section 3.1.2) and strictly enlarge feasible support under a token budget (Section 3.2).

**Search and Retrieval Agents.** Consider web search, retrieval APIs, or domain databases (e.g., scholarly indices, code search). Let an external retriever implement a (possibly stochastic) mapping $\mathcal{R} : (q, s) \mapsto r$, where $q$ is a query issued by the LLM and $s$ is the (latent) world/index state at the time of the call. Even when $\mathcal{R}$ is not perfectly deterministic, the *trajectory* that includes the returned snippet $r$ is unreachable for a pure-text model unless it *guesses* the salient facts in $r$ token-by-token. This mirrors the random-oracle argument in Theorem 3.4: as the entropy of $r$ conditioned on $(q, x)$ grows, the probability that a pure-text model reproduces $r$ by chance decays exponentially, while a tool-augmented model obtains $r$ via a single call. Hence support expands, and under any fixed budget $B$ the feasible set also strictly expands once the text-only paraphrase of $r$ would exceed $B$.

**Checkers, Verifiers, and Program Runners.** Beyond "heavy" computation, many tools act as *verifiers*: unit tests, symbolic algebra checkers, SAT/SMT solvers, theorem provers, type checkers, or even a Python REPL used only to validate a candidate answer. Such tools add *deterministic pruning* transitions to the trajectory graph: incorrect branches are cut immediately with $O(1)$ tokens. This reduces the exploration burden under RLVR-style training and enlarges the set of practically reachable strategies under a budget.

**Stateful External Memory.** Tools can expose memory larger and more persistent than the model's context: key–value caches, external scratchpads, vector stores, or file systems. Each call updates an external state $m_{t+1} = U(m_t, a_t)$ and reads views $v_t = V(m_t)$ at $O(1)$ token cost. Strategies that require memory $|m| \gg B$ are impossible to realize faithfully in pure text (which must inline $m$), but become feasible when memory lives outside the context window.

**Proposition E.1** (Informal; External State as Unbounded Scratchpad). *Suppose an algorithm requires* $\Omega(n)$ *writable memory cells for problem size* $n$. *If a tool exposes these cells with per-step I/O* $O(1)$, *then for sufficiently large* $n$, *the algorithm's equivalence class lies in* $\text{supp}_B(p_{TIR})$ *but not in* $\text{supp}_B(q_{text})$ *for any fixed* $B$.

**Embodied and Interactive Environments.** When the LLM acts in an MDP or game environment (Xu et al., 2025), the environment transition $s_{t+1} = E(s_t, a_t)$ is itself an *external oracle*. Our earlier support-expansion argument applies verbatim: trajectories that include specific environment observations or states are unreachable by text-only generation unless they are guessed token-by-token. Token-efficiency arguments also lift: environment interactions can realize long-horizon plans with *summarized* textual traces, whereas a pure-text simulation would require enumerating each counterfactual step.

**Noisy or Non-Deterministic Tools.** Stochastic returns (e.g., fluctuating search rankings) do not invalidate support expansion. What matters is the existence of *some* positive-probability outputs with substantial conditional entropy that are infeasible to reproduce via text within budget. In other words, determinism is a convenience, not a necessity, for our conclusions.

**Composing Multiple Tools.** Real agents chain retrieval, computation, verification, and environment actions. Composition behaves monotonically:

**Proposition E.2** (Informal; Monotone Closure under Composition). *Let* $\mathcal{T}_1, \ldots, \mathcal{T}_k$ *be tools with per-call costs that sum to at most* $B$. *If each* $\mathcal{T}_i$ *individually yields a strict feasible-support gain for some subproblem family at size* $n_i$, *then there exist composite tasks for which the sequential (or branched) use of* $\{\mathcal{T}_i\}$ *yields a strict feasible-support gain over any pure-text policy at the same total budget.*

**Takeaway.** "Python" is merely one instantiation of a broader principle, our extensions unify code execution, search, verification, memory, and embodied interaction under the same analytical lens.

## F    EXAMPLES ON TOKEN EFFICIENCY

Table 1: Contrasting Token Efficiency for an Iterative Task ($N \to \infty$)

| Programmatic Approach (Python) | Natural Language Reasoning |
|---|---|
| A symbolic, abstract representation of the computation. The token cost is constant and independent of $N$. | A concrete, step-by-step enumeration of the computation. The token cost scales with the magnitude of $N$. |
| 
```python
# N can be 10,000,000 or more
for i in range(N):
    # Perform some check
    check(i)

```
 | *"Okay, to solve this, I must check every number.* *First, for n=1, I perform the check...* *Next, for n=2, I perform the check...* *Next, for n=3, I perform the check...* *...* *(This enumeration continues for millions of steps)* *...* *Finally, for n=10,000,000, I perform the check..."* |
| **Token Cost:** A few dozen tokens. Scales as $O(1)$. This is highly efficient and scalable. | **Token Cost:** Proportional to $N$. Scales as $\Omega(N)$. This is inefficient and becomes intractable for large $N$, quickly exceeding any feasible context window. |

Table 2: Contrasting Token Efficiency for Solving Large Linear Systems

| Programmatic Approach (Python) | Natural Language Reasoning |
|---|---|
| A single call to a highly optimized numerical library solves $Ax = b$. The token cost is constant, independent of the matrix dimension $n$. | A detailed explanation of Gaussian elimination, requiring a description of each row operation. The token cost scales with the matrix size. |
| 
```python
import numpy as np
# A is a large n x n matrix,
# e.g., n=1000
x = np.linalg.solve(A, b)
```
 | *"To solve the system, we perform Gaussian elimination. First, to eliminate the first variable from the second row, we subtract $A_{2,1}/A_{1,1}$ times the first row from the second row. We must do this for all $n-1$ rows below the first. Next, we use the new second row to eliminate the second variable from the rows below it... (This narration continues for $O(n^2)$ elements and $O(n^3)$ operations)."* |
| **Token Cost:** A few tokens. Scales as $O(1)$. Enables solving massive systems within a tiny token budget. | **Token Cost:** Proportional to the number of elements in the matrix to sketch. Scales as $\Omega(n^2)$. A full narration would scale as $\Omega(n^3)$. |

Table 3: Contrasting Token Efficiency for a Dynamic Programming Task (Fibonacci Sequence)

| Programmatic Approach (Python) | Natural Language Reasoning |
|---|---|
| A compact representation of the recurrence relation, with a token cost independent of the input integer $N$. | A verbose, step-by-step calculation of every sub-problem's solution, with a token cost that grows with $N$. |

| | |
|---|---|
| ```python
memo = {0: 0, 1: 1}
def fib(n):
  if n in memo: return memo[n]
  memo[n] = fib(n-1) + fib(n-2)
  return memo[n]
``` | *"To get fib(5), I need fib(4) and fib(3). Fib(2) is fib(1)+fib(0) = 1+0 = 1. Fib(3) is fib(2)+fib(1) = 1+1 = 2. Fib(4) is fib(3)+fib(2) = 3+1 = 4. So, fib(5) is fib(4)+fib(3) = 4+2 = 6... Wait, let me recheck. fib(4) is 3+2=5. No, fib(4) is 2+1=3. Okay, so fib(5) is 3+2=5."* |
| **Token Cost:** $O(1)$ | **Token Cost:** $\Omega(N)$ |

Table 4: Contrasting Token Efficiency for Search Algorithms

| Programmatic Approach (Python) | Natural Language Reasoning |
|---|---|
| An abstract procedure for state-space traversal, using data structures like a queue and a set. | A full, step-by-step narration of the entire exploration process, including every node visited and every state change of the queue. |

| | |
|---|---|
| ```python
from collections import deque

def bfs(graph, start_node):
  queue = deque([start_node])
  visited = {start_node}
  while queue:
    node = queue.popleft()
    # Process node
    for neighbor in graph[node]:
      if neighbor not in visited:
        visited.add(neighbor)
        queue.append(neighbor)
``` | *"I start at node 'A'. Queue is ['A'], visited is 'A'. I pop 'A'. Its neighbors are 'B', 'C'. Queue is now ['B', 'C'], visited is 'A','B','C'. I pop 'B'. Its neighbor is 'D'. Queue is now ['C', 'D'], visited is 'A','B','C','D'. I pop 'C'..." (and so on)* |
| **Token Cost:** Constant cost for the algorithm's definition. Scales as $O(1)$. | **Token Cost:** Proportional to the number of vertices and edges, $V + E$. Scales as $\Omega(V + E)$. |

## G ANALYSIS OF THE FAILED REWARD-BASED APPROACH

To encourage the earlier code invocation, our initial and most direct approach was to introduce an *early-code reward* directly into the reward function. For each response $i$ that is both correct and code-containing in a group of samples, we added a reward term $r'_i$ that penalizes later code invocation:

$$R_i = 1 + r'_i \quad \text{where} \quad r'_i = \delta \cdot \text{clip}\left(\frac{p_i - \text{mean}(\mathbf{p})}{\text{std}(\mathbf{p})}, -c, c\right).$$

where $\mathbf{p}$ is the set of first code invocation positions for all correct, code-containing responses within the group. Furthermore, $\delta$ is a negative coefficient to encourage early code invocation, and $c$ is a clipping hyperparameter. However, this seemingly innocuous modification proved to be highly destabilizing during training (see experimental details in Section 4.4 and Figure 3 (a)). In algorithms like GRPO that rely on group normalized advantage, this design has a critical flaw. In the common scenario where all samples in a group are correct, the primary reward signal (the constant '1') is

entirely eliminated by the normalization. The advantage calculation then becomes:

$$A_i = \frac{R_i - \text{mean}(\mathbf{R})}{\text{std}(\mathbf{R})} = \frac{(1 + r_i') - (1 + \text{mean}(\mathbf{r}'))}{\text{std}(\mathbf{r}')} = \frac{r_i' - \text{mean}(\mathbf{r}')}{\text{std}(\mathbf{r}')}$$

This leads to a catastrophic outcome: (1) the primary signal about answer correctness disappears, (2) the auxiliary signal $r_i'$ is amplified to the same magnitude as the original primary signal, and (3) due to the nature of standardization, approximately half of these correct responses receive a negative advantage and are thus heavily penalized, solely because their code invocation is later than the group's average.

## H EXPERIMENTAL SETUP

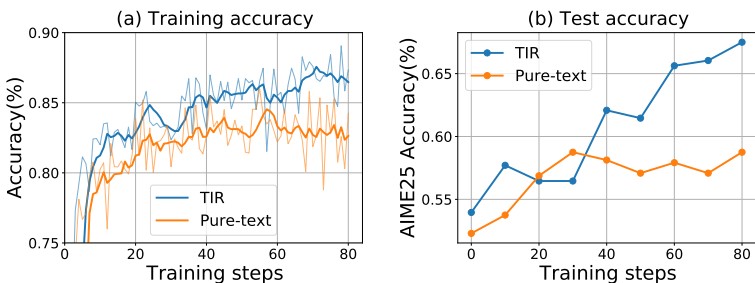

Figure 5: The (a) training and (b) testing accuracy of the TIR and pure-text RL on Qwen3-8B model. The AIME25 accuracy (b) is the average of 16 responses.

**Model and Datasets.** All experiments are based on the Qwen3-8B model (Qwen, 2025). For our training data, we randomly sample 10,000 English problems from the DAPO dataset (Yu et al., 2025) due to limited computational resources. Since our aim is to fundamentally understand the mechanisms of TIR *rather than* to improve absolute accuracy of benchmarks, this dataset is sufficient for our purpose, in contrast to the extensive training datasets used in other literature (Yu et al., 2025; Feng et al., 2025). Our primary evaluation benchmarks are AIME24, AIME25, and a challenging subset of the Omni-MATH dataset (Gao et al., 2024). For the latter, due to the large size of the dataset, we curated the 512 *most difficult* problems that are amenable to reliable, rule-based evaluation, which we denote as Omni-MATH-512. While AIME-style problems have numerical final answers, Omni-MATH-512 contains a substantial fraction of problems whose final answers are symbolic expressions rather than single numbers. Together with the large answer space (000–999 for AIME, symbolic expressions for part of Omni-MATH-512), this makes blind guessing an implausible explanation for the observed gains.

**Training Protocol.** We train two main models for comparison: our proposed TIR model, which can execute code to assist in its reasoning process, and a pure-text RL model as a baseline (as shown in Figure 5). Both models are trained for 3 epochs using the DAPO algorithm (Yu et al., 2025), a variant of GRPO (DeepSeek, 2025), with the same dataset and hyperparameters; the only difference is an additional system prompt that specifies the tool usage format for the TIR model. During training, we use a rollout batch size of 96 problems, with 8 responses sampled per problem, a maximum response length of 16,384 tokens, and a sampling temperature of 1.0 to encourage exploration.

**Evaluation Protocol.** For evaluations, we set the sampling temperature to 0.6 and maximum response length to 16,384 tokens unless otherwise specified.

To avoid overclaiming, we explicitly clarify the scope of our experimental setup. Our experiments focus on a single tool (Python interpreter), a single model family (Qwen3-8B), and mathematical reasoning tasks. This design choice is motivated by the goal of isolating and interpreting the core mechanisms of Tool-Integrated Reasoning, rather than benchmarking a broad set of tools or domains. Python provides a deterministic, high–information-density computational interface that allows us to cleanly analyze support expansion. Although we do not empirically test other tools, Appendix E discusses how our theoretical arguments extend to a broad family of external tools. We leave multi-tool, multi-domain evaluations to future work.

# I  PASS@$k$ DATA

Table 5 shows the detailed pass@$k$ results for the TIR and pure-text models across the three benchmarks, evaluated with the max sample size of 256.

Table 5: Pass@$k$ results for the TIR model and the pure text model

| k | AIME24 | | AIME25 | | Omni-MATH-512 | |
|---|---|---|---|---|---|---|
| | **TIR** | **Pure Text** | **TIR** | **Pure Text** | **TIR** | **Pure Text** |
| 1 | 0.7829 | 0.6331 | 0.6841 | 0.5184 | 0.5128 | 0.3585 |
| 2 | 0.8408 | 0.7184 | 0.7818 | 0.6065 | 0.5885 | 0.4208 |
| 4 | 0.8632 | 0.7703 | 0.8395 | 0.6730 | 0.6437 | 0.4707 |
| 8 | 0.8825 | 0.8050 | 0.8792 | 0.7262 | 0.6869 | 0.5153 |
| 16 | 0.9024 | 0.8312 | 0.9117 | 0.7613 | 0.7232 | 0.5570 |
| 32 | 0.9173 | 0.8496 | 0.9339 | 0.7810 | 0.7545 | 0.5942 |
| 64 | 0.9312 | 0.8645 | 0.9503 | 0.7979 | 0.7802 | 0.6271 |
| 128 | 0.9480 | 0.8813 | 0.9625 | 0.8250 | 0.8018 | 0.6575 |
| 256 | 0.9667 | 0.9000 | 0.9667 | 0.8667 | 0.8203 | 0.6836 |

# J  TEMPERATURE SENSITIVITY ANALYSIS

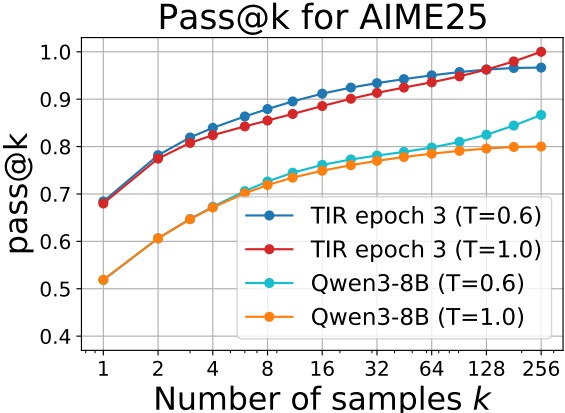

Figure 6: Pass@$k$ curves on AIME25 under temperatures 0.6 and 1.0 for the TIR (RL trained) and pure-text models (Qwen3-8B) model.

To examine whether our conclusions depend on sampling temperature, we measured pass@$k$ on AIME25 under temperatures 0.6 and 1.0 for both TIR and pure-text model, as shown in Figure 6. For small k, both models show almost identical pass@k curves across temperatures. For large k, increasing the temperature to 1.0 increases the diversity of TIR trajectories and slightly improves its tail-end pass@k. In contrast, the pure-text model becomes noticeably noisier under temperature 1.0, resulting in lower pass@256. This phenomenon is consistent with prior observations in Yue et al. (2025), which reports that the base model drops but the RL-trained model remains stable when temperature exceeds 1.0.

Across temperatures, the TIR model consistently outperforms the pure-text model. This supports our core claim that TIR's gains stem from structural support expansion, rather than sampling stochasticity.

## K CAPABILITY EXPANSION AND SHRINKAGE

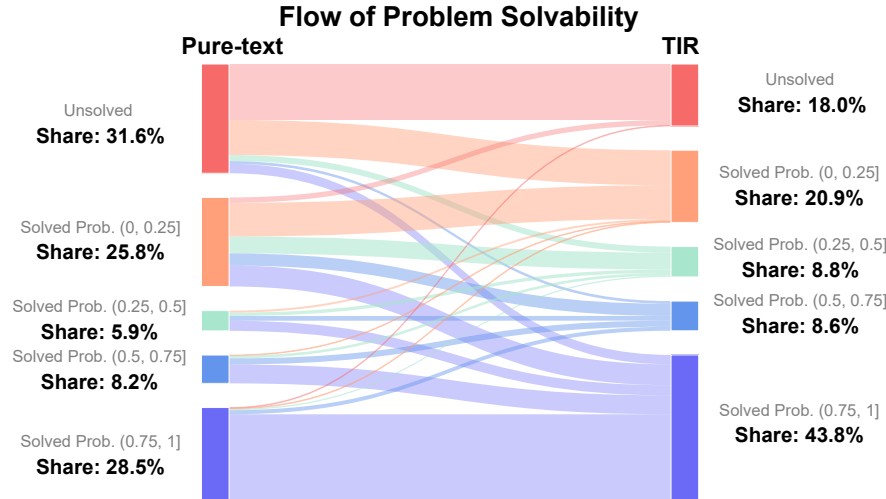

Figure 7: The detailed flow of problem solvability on Omni-MATH-512 when transitioning from the pure-text model to the TIR model. The solved probability of each problem is evaluated at $k = 256$.

## L ANALYSIS OF THE $1.8\%$ CAPABILITY SHRINKAGE

To better understand the small "capability shrinkage" observed in Figure 2, where the TIR model fails on 9 problems that the pure-text model solves, we conducted a detailed item-level analysis. Across 512 evaluated problems, shrinkage occurs in 9 cases (1.8%), which is substantially smaller than the gain in newly solved problems (+15.4%). Below we show that the shrinkage is either attributable to sampling variance or to RL training artifacts, rather than an inherent limitation of TIR.

**1.** For 6 of the 9 problems, the pure-text model solved them only 1 or 2 times out of 256 samples, while the TIR model solved them 0 times. These problems have extremely low base success probabilities, and a shift from 1–2 correct samples to 0 correct samples is fully explainable by Monte-Carlo variance. So these cases do not reflect a meaningful capability shrinkage.

**2.** The remaining 3 problems exhibit larger drops. To determine whether this is caused by TIR or by RL training, we evaluated a non-RL TIR model (vanilla Qwen3-8B + Python tool, no RL fine-tuning) on these 3 problems. This model preserved the pure-text model's accuracy on all 3 problems, indicating that the drops were introduced by RL, not by the TIR mechanism itself. This aligns with prior observations that RL can occasionally induce forgetting and cause capability shrinkage (Yue et al., 2025; Wu et al., 2025).

Thus, the 1.8% shrinkage should be interpreted as: A small, explainable artifact of RL fine-tuning and sampling variance, rather than a limitation of TIR. This complements our main result that TIR achieves large support expansion (+15.4% newly solved problems), far outweighing these minor, non-inherent shrinkage effects.

## M    RUBRIC FOR ALGORITHMIC FRIENDLINESS

Table 6 shows the rubric we use for Gemini Pro APIs (Gemini, 2025) to classify the math problems.

Table 6: Rubric for assessing the "algorithmic friendliness" of problems.

| Score | Level | Description | Required Insight |
|-------|-------|-------------|------------------|
| 5 | **Very High** (Direct Application) | The problem is a textbook example for a standard algorithm (e.g., backtracking). The problem statement itself almost serves as the specification. **Almost no mathematical insight is needed**. | None beyond basic arithmetic. |
| 4 | **High** (Minor Insight) | An algorithm provides a clear advantage, but requires a **standard, well-known mathematical identity** or **simple transformation** to be applied. The mathematical hurdle is low. | Recalling and applying a common formula or theorem. |
| 3 | **Medium** (Significant Insight) | A computational solution is effective, but only after applying a **significant mathematical insight** or performing **complex problem modeling**. The difficulty is substantial. | A creative, problem-specific trick or a complex modeling effort. |
| 2 | **Low** (Impractical Algorithm) | An algorithm is theoretically possible but highly impractical (enormous search space, precision issues). The algorithmic optimizations are **equivalent in difficulty to the mathematical solution**. | Insights needed are essentially the mathematical solution itself. |
| 1 | **Very Low** (Non-computational) | The problem is fundamentally abstract and cannot be solved by computation (e.g., requires a formal proof, deals with uncountable sets). | N/A. |

# N    PASS@$k$ CURVES FOR PROBLEMS GROUPED BY ALGO FRIENDLINESS

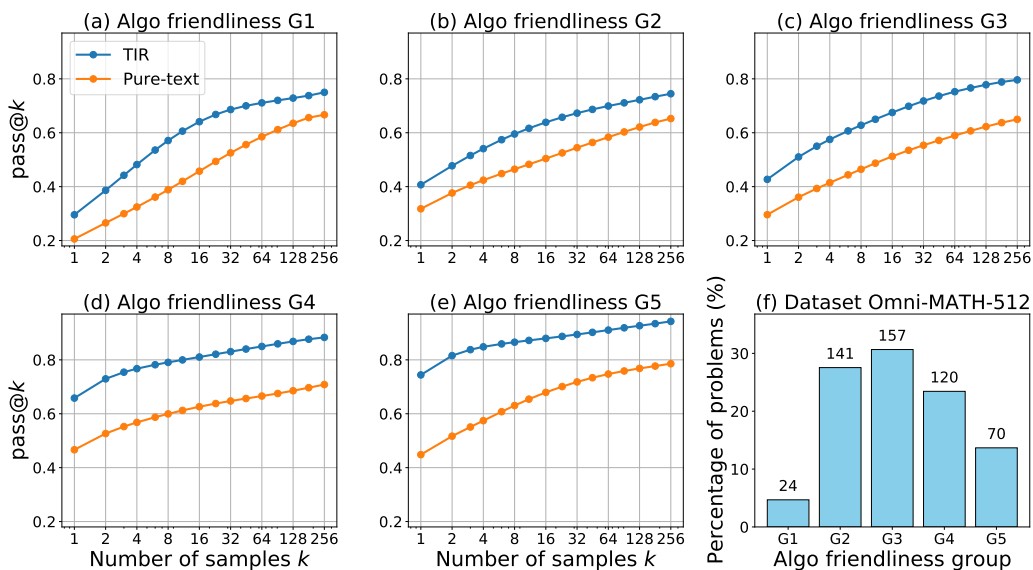

Figure 8:  (a)-(e) Pass@k curves for the TIR and pure-text models, grouped by problem algo friendliness. (f) The distribution of algo friendliness scores across the Omni-MATH-512 dataset. The problems are categorized into five groups based on their algo friendliness scores: 1.0–1.5 (G1), 2.0–2.5 (G2), 3.0–3.5 (G3), 4.0–4.5 (G4), and 5.0 (G5).

## O  Emergent Cognitive Patterns of Tool Integration

The quantitative results of the previous sections demonstrate *that* TIR is universally effective, but they do not fully explain *how*. If the model's advantage is not limited to algorithmically amenable problems, how exactly is it leveraging the code interpreter to solve problems requiring abstract insight? Through qualitative analysis of model outputs, we have identified three distinct and recurring patterns of code utilization that answer this question.

**Pattern 1: Insight-to-computation transformation.** In this primary pattern, the model's first step is not to code, but to *reason*. It engages in non-trivial, text-based analysis to deconstruct a complex problem, applying mathematical insights to transform it into a state that is amenable to a programmatic solution. The code interpreter is then invoked to execute a genuine algorithm (such as search, enumeration, or DP) that efficiently solves this newly formulated sub-problem under a limited computational resource. Unlike straightforward sequential calculations one might perform with a calculator, these algorithms often possess complex control flows (e.g., loops, recursion) that are challenging for a language model to emulate or follow step-by-step.

For instance, as shown in Table 7, the model first uses mathematical reasoning to derive a transcendental equation from the abstract geometric problem. It then employs code to iterate the entire parameter space of $(m, n)$ pairs, using the Intermediate Value Theorem as a numerical method to efficiently detect whether a solution exists for each pair.

Table 7: A case of insight-to-computation transformation.

---

**[Question]**
Let $a \neq b$ be positive real numbers and $m, n$ be positive integers. An $m + n$-gon $P$ has the property that $m$ sides have length $a$ and $n$ sides have length $b$. Further suppose that $P$ can be inscribed in a circle of radius $a + b$. Compute the number of ordered pairs $(m, n)$, with $m, n \leq 100$, for which such a polygon $P$ exists for some distinct values of $a$ and $b$.

---

**[Response]**
Okay, let's try to tackle this problem step by step.
…
We need to find the number of ordered pairs $(m, n)$ where $m, n \leq 100$ such that this equation has solutions for $a$ and $b$ with $a \neq b$. … So, substituting into the equation:

$$m \cdot \arcsin(x/2) + n \cdot \arcsin((1 - x)/2) = \pi$$

…
But how can I find all such $(m, n)$ pairs? Since $m$ and $n$ are up to 100, perhaps we can write a program to check for each possible pair $(m, n)$ whether there exists an $x$ in $(0, 1)$ such that the equation is satisfied. But how to implement this check in Python? The problem is that the equation is transcendental and might not have an analytical solution, so numerical methods would be needed. For each $(m, n)$, we can define a function $f(x) = m \cdot \arcsin(x/2) + n \cdot \arcsin((1 - x)/2) - \pi$, and check if there's a root of $f(x) = 0$ in $(0, 1)$, excluding $x = 0.5$. But how to check for roots? For example, we can use the Intermediate Value Theorem. If $f(x)$ changes sign over an interval, there's a root in that interval.
…
Now, let's proceed to write the Python code.



```
import math

def has_solution(m, n):
  def f(x):
    return m * math.asin(x/2) + n * math.asin((1 - x)/2) - math.pi

  # Check interval (0, 0.5)
  left = 0.0001
  right = 0.4999
  if f(left) * f(right) < 0:
    return True

  # Check interval (0.5, 1)
```

```
14    left = 0.5001
15    right = 0.9999
16    if f(left) * f(right) < 0:
17        return True
18
19    return False
20
21  count = 0
22  for m in range(1, 101):
23    for n in range(1, 101):
24        if has_solution(m, n):
25            count += 1
26
27  print(count)
28
```



<interpreter> 940 </interpreter>

...

Wait, but the code is written in Python, and the interpreter output is 940. So the final answer is $\boxed{940}$.

**Pattern 2: Exploration and verification via code.** For problems where the solution path is not immediately obvious, the model utilizes the code interpreter as an interactive sandbox for exploration and hypothesis testing. Instead of committing to a *single line* of reasoning, it formulates conjectures and writes short code snippets to test them, observe their outcomes, and iteratively refine its strategy based on the feedback. This pattern is particularly prevalent in problems with low algorithmic amenability, where it allows the model to build confidence and discover insights through empirical experimentation.

Table 8 provides a clear instance of pattern 2: exploration and verification via code. The model first derives a candidate value of $\lambda = \sqrt{3}$ from a simple case, then uses the code interpreter to numerically explore more different scenarios. The feedbacks validate its initial hypothesis and pivot its strategy from further exploration toward constructing a rigorous algebraic proof.

These first two patterns represent a fundamental departure from pure-text reasoning. As we established in Section 3, they constitute entirely new *Computational Equivalence Classes*, new strategies for solving problems. While a pure-text model might theoretically be able to simulate these processes, the token cost of doing so would be astronomical. The step-by-step, trial-and-error nature of the exploratory pattern, in particular, would lead to a blow-up in token length. Therefore, these strategies lie far outside the *Feasible Support under Budget B* for any practical context window, making them uniquely accessible to the TIR paradigm.

Table 8: A case of exploration and verification via code.

**[Question]**
Find the largest real number $\lambda$ with the following property: for any positive real numbers $p, q, r, s$ there exists a complex number $z = a + bi (a, b \in \mathbb{R})$ such that

$$|b| \geq \lambda|a| \quad \text{and} \quad (pz^3 + 2qz^2 + 2rz + s) \cdot (qz^3 + 2pz^2 + 2sz + r) = 0.$$

**[Response]**
Okay, let's try to tackle this problem step by step. . . . . .
But how do we approach this? Maybe we can think about the problem by considering specific cases . . . Perhaps the worst case occurs when $p = q$ and $r = s$? . . . Let's try to consider a symmetric case. . . . Maybe choosing $p = q = 1$ and $r = s = 1$. Let's try that. . . .
The roots are $z = -1$ and roots of $z^2 + z + 1 = 0$, which are $\frac{-1 \pm i\sqrt{3}}{2}$. Let's compute their $|b|/|a|$:

$$z = \frac{-1 \pm i\sqrt{3}}{2} \rightarrow a = -1/2, b = \pm\sqrt{3}/2.$$

So $|b| = \sqrt{3}/2$, $|a| = 1/2$. So $|b|/|a|$ is $\sqrt{3}$.

So for this case, $\lambda$ can be as large as $\sqrt{3}$. But this is just for this specific case. ...

This is getting a bit abstract. Maybe we can consider a case where both $P(z)$ and $Q(z)$ have roots with minimal $|b|/|a|$, and find what that value is. ... when $t$ is very small. Let me try to find its roots numerically for a specific small $t$. Let me write a Python code snippet to find the roots of $P(z) = z^3 + 2z^2 + 0.2z + 0.1$.



```
import numpy as np

# Coefficients for P(z) = z^3 + 2z^2 + 0.2z + 0.1
coeffs = [1, 2, 0.2, 0.1]
roots = np.roots(coeffs)
print(roots)

```



<interpreter> [-1.92303902+0.j,     -0.03848049+0.22476716j,     -0.03848049-0.22476716j]

</interpreter>

For these complex roots, $|b|/|a|$ is $0.2248/0.0385 \approx 5.83$. ... Wait, but this case gives a high value of $|b|/|a|$. So maybe the worst case is when the minimal such value is low. ...

But how to find that? It seems challenging without more insight. ... Alternatively, consider using symmetry. ... But if we set $s = 4qr/p$, for example, let's set $p = 1, q = 1, r = 1$, then $s = 4 * 1 * 1/1 = 4$. So $P(z) = z^3 + 2z^2 + 2z + 4$. Let's find its roots. Using numpy again:



```
import numpy as np

coeffs = [1, 2, 2, 4]
roots = np.roots(coeffs)
print(roots)

```



<interpreter> [-2.00000000e+00+0.j,     1.66533454e-16+1.41421356j,     1.66533454e-16-1.41421356j]

</interpreter>

So $|b|/|a|$ is infinite (since $a$ is nearly zero), but for practical purposes, this suggests that for this case, there are roots with very large $|b|/|a|$. So this case is not helpful. ... This suggests that for this case, we can achieve $|b|/|a|$ up to $\sqrt{3}$.

...

Let's think of a case where the ratio is lower than $\sqrt{3}$. ... Let me consider the case where $p = q = 1, r = 1, s = 2$. Then $P(z) = z^3 + 2z^2 + 2z + 2$. Let's find its roots using numpy.



```
import numpy as np

coeffs = [1, 2, 2, 2]
roots = np.roots(coeffs)
print(roots)

```



<interpreter> [-1.54368901+0.j,     -0.22815549+1.11514251j,     -0.22815549-1.11514251j]

</interpreter>

So $|b|/|a| = 1.115/0.228 \approx 4.89$. The real root has $|b|/|a| = 0$. So again, the maximum is 4.89.

This doesn't seem to help. ... Let me think of a different approach. ...

Let me assume that there is a root $z = a + bi$ with $|b| = \lambda|a|$. Let's assume $a \neq 0$. Then, $b = \lambda a$ or $b = -\lambda a$. Let's assume $b = \lambda a$. ... Substitute $z = a + \lambda ai = a(1 + \lambda i)$ into $P(z) = pz^3 + 2qz^2 + 2rz + s = 0$.

...

Since $a \neq 0$ and $\lambda$ is non-zero (otherwise, we have real roots), we can divide both sides by $\lambda a$:

$$pa^2(3 - \lambda^2) + 4qa + 2r = 0$$

This is one equation.

From the equation, since all terms except the first are positive, we must have $(3 - \lambda^2) < 0$, which implies $\lambda^2 > 3$, or $\lambda > \sqrt{3}$. This suggests that for *any* non-real root, the ratio $|b|/|a|$ must be greater than $\sqrt{3}$.

...

So, we have two key pieces of information:

1. We found a specific case ($p = q = r = s = 1$) where the largest available ratio is exactly $\sqrt{3}$. This means $\lambda$ cannot be greater than $\sqrt{3}$.

2. Our general proof shows that for any case with a non-real root, the ratio must be greater than or equal to $\sqrt{3}$. This means $\lambda$ must be at least $\sqrt{3}$.

Combining these, the largest possible value for $\lambda$ is exactly $\boxed{\sqrt{3}}$.

---

**Pattern 3: Offloading complex calculation.** This is the most direct pattern of tool use, where the model has a clear, linear path to the solution but delegates complex or tedious calculations to the interpreter. This usage aligns with the naive view of TIR as a "calculator", but its importance should not be understated. By offloading these steps, the model minimizes the *risk* of unforced computational errors that frequently derail long chains of pure-text thought, thereby preserving the integrity of the overall reasoning process.

A representative example is shown in Table 9. Here, the model first performs the text-based reasoning to establish a solution path, then uses the interpreter as a precision tool to execute the series of intricate vector and algebraic computations that would be highly prone to manual error.

In conclusion, these emergent patterns reveal a sophisticated interplay between the LLM's reasoning capabilities and the code interpreter's computational power. The model is not merely using a tool; it is *thinking with* tools. This signifies a fundamental shift in strategy: rather than simply delegating calculations from an otherwise unchanged, text-based line of thought, the model learns to generate novel problem-solving approaches that are intrinsically synergistic with the interpreter. It formulates plans that leverage programmatic strengths like iteration and DP from the outset, developing new "Computational Equivalence Classes" that were previously inaccessible. Such dynamic and flexible code invocation enables the TIR model to break the capability ceiling of its pure-text counterpart.

Table 9: A case of offloading complex calculation.

---

**[Question]**

Let acute triangle $ABC$ have circumcenter $O$, and let $M$ be the midpoint of $BC$. Let $P$ be the unique point such that $\angle BAP = \angle CAM$, $\angle CAP = \angle BAM$, and $\angle APO = 90°$. If $AO = 53$, $OM = 28$, and $AM = 75$, compute the perimeter of $\triangle BPC$.

---

**[Response]**

Okay, let's try to solve this problem step by step.

...

Alternatively, maybe using coordinate geometry. Let me think about setting up coordinates.

...

First, compute $A_x$, $A_y$ numerically.

$$A_x = (75\sqrt{13})/7, \qquad A_y = 450/7$$

Then compute vectors AB and AC, their lengths, unit vectors, angle bisector direction, then reflect AM over this direction to get the direction of AP. Then find the intersection of this line with the circle.

But implementing all this in code is time-consuming, but let's try. Let me write the code.



```
import math

# Given values
AO = 53
```

```
 5  OM = 28
 6  AM = 75
 7
 8  # Compute Ay
 9  Ay = 450 / 7
10  # Compute Ax
11  sqrt13 = math.sqrt(13)
12  Ax = (75 * sqrt13) / 7
13
14  # Coordinates of A
15  A = (Ax, Ay)
16  # Coordinates of O
17  O = (0, 28)
18  # Coordinates of M
19  M = (0, 0)
20  # Coordinates of B and C
21  B = (-45, 0)
22  C = (45, 0)
23
24  # Function to compute vector from point1 to point2
25  def vector(p1, p2):
26      return (p2[0] - p1[0], p2[1] - p1[1])
27
28  # Compute vectors AB and AC
29  AB = vector(A, B)
30  AC = vector(A, C)
31
32  # Compute lengths of AB and AC
33  len_AB = math.hypot(AB[0], AB[1])
34  len_AC = math.hypot(AC[0], AC[1])
35
36  # Compute unit vectors
37  unit_AB = (AB[0]/len_AB, AB[1]/len_AB)
38  unit_AC = (AC[0]/len_AC, AC[1]/len_AC)
39
40  # Angle bisector direction
41  angle_bisector_dir = (unit_AB[0] + unit_AC[0], unit_AB[1] + unit_AC
        [1])
42
43  # Vector AM
44  AM_vec = vector(A, M)
45
46  # Function to reflect a vector over another vector
47  def reflect_vector(v, axis):
48      # Normalize the axis vector
49      axis_len = math.hypot(axis[0], axis[1])
50      if axis_len == 0:
51          return v
52      axis_unit = (axis[0]/axis_len, axis[1]/axis_len)
53
54      # Dot product
55      dot = v[0] * axis_unit[0] + v[1] * axis_unit[1]
56
57      # Projection of v onto axis
58      proj = (dot * axis_unit[0], dot * axis_unit[1])
59
60      # Reflection
61      reflected = (2*dot*axis_unit[0] - v[0], 2*dot*axis_unit[1] - v[1])
62      return reflected
63
64  # Reflect AM_vec over angle bisector direction
65  AP_dir = reflect_vector(AM_vec, angle_bisector_dir)
66
```

```python
# Normalize AP_dir to get direction
AP_dir_len = math.hypot(AP_dir[0], AP_dir[1])
AP_dir_unit = (AP_dir[0]/AP_dir_len, AP_dir[1]/AP_dir_len)

# Parametric line for AP: A + t * AP_dir_unit
# Find intersection with circle with diameter AO

# Equation of circle with diameter AO
# (x - mid_x)^2 + (y - mid_y)^2 = r^2
mid_x = (A[0] + O[0])/2
mid_y = (A[1] + O[1])/2
r = AO / 2

# Parametric equations for line AP
# x = A[0] + t * AP_dir_unit[0]
# y = A[1] + t * AP_dir_unit[1]

# Substitute into circle equation
# (x - mid_x)^2 + (y - mid_y)^2 = r^2
# Expand and solve for t

def solve_for_t():
    # x = A[0] + t * dx
    # y = A[1] + t * dy
    dx = AP_dir_unit[0]
    dy = AP_dir_unit[1]

    # Expand (x - mid_x)^2 + (y - mid_y)^2 = r^2
    # (A[0] + t*dx - mid_x)^2 + (A[1] + t*dy - mid_y)^2 = r^2
    # Expand terms
    a = dx**2 + dy**2
    b = 2 * (dx*(A[0] - mid_x) + dy*(A[1] - mid_y))
    c = (A[0] - mid_x)**2 + (A[1] - mid_y)**2 - r**2

    # Solve a*t^2 + b*t + c = 0
    discriminant = b**2 - 4*a*c
    if discriminant < 0:
        return None
    t1 = (-b + math.sqrt(discriminant)) / (2*a)
    t2 = (-b - math.sqrt(discriminant)) / (2*a)
    return t1, t2

t_values = solve_for_t()
if t_values:
    t = t_values[0]  # Choose one solution, the other might be A itself
    px = A[0] + t * AP_dir_unit[0]
    py = A[1] + t * AP_dir_unit[1]
    P = (px, py)
else:
    P = None

# Now compute perimeter of triangle BPC
if P:
    def distance(p1, p2):
        return math.hypot(p1[0]-p2[0], p1[1]-p2[1])

    BP = distance(B, P)
    CP = distance(C, P)
    BC = distance(B, C)
    perimeter = BP + CP + BC
    print(perimeter)
else:
    print("No intersection found")
```

130



<interpreter> 192.0 </interpreter>

The code computes the perimeter of triangle BPC and returns 192.0. Therefore, the answer is 192 .

