# OpenReview forum: "Understanding Tool-Integrated Reasoning"
_ICLR.cc/2026/Conference — Submitted to ICLR 2026_

### Official Review · Reviewer_zSw2 · 2025-10-28

**Soundness:** 2
**Presentation:** 3
**Contribution:** 2
**Rating:** 4
**Confidence:** 3

**Summary:**

This paper examines tool-integrated reasoning (TIR) for LLMs and introduces ASPO, a training tweak that adds a clipped/normalized term to the advantage to prefer earlier and shorter tool calls (e.g., code execution, verifier). The theoretical sections frame tools as expanding both the support set (what solutions are reachable at all) and the feasible support set under finite token budgets, arguing that external computation can compress long textual derivations into short calls, thereby enlarging reachable solutions without increasing sequence length. The algorithmic claim is that naive reward-level shaping is unstable in GRPO-like pipelines, whereas advantage-level shaping (ASPO) is simpler and more stable in practice. Empirically, the paper reports pass@k gains on math benchmarks (AIME24 AIME 25, Omni-MATH), alongside qualitative evidence that policies learn to invoke tools earlier.

**Strengths:**

1. Understanding when/why tools help LLM reasoning is important and practically relevant.
2. The support set / feasible support set expansion lens offers an intuitive way to talk about why tools could enlarge reachable solutions under token budgets.
3. Improvements are reported across several math-reasoning benchmarks, suggesting the approach is not dataset-specific.

**Weaknesses:**

1. Although the paper positions its theoretical proofs as a primary contribution (as stated in the Contributions), the core results largely repackage a self-evident intuition: if a deterministic tool can collapse a long derivation into a one-step call, its reachable set will strictly dominate text-only generation. The theorems read more like formal wrapping of an intuition than analysis that meaningfully narrows uncertainty about real systems.
2.  ASPO modifies the advantage with a clipping/normalization term tied to “earliness” and tool length. Unlike potential-based shaping (which preserves optimal policies), this construction can change the fixed point of policy optimization. Will it incentivize spurious early tool calls in tasks that do not need tools?

**Questions:**

See my comments on weaknesses

---

> ### Author Response · Authors · 2025-11-26
> **Response to Reviewer zSw2**
>
> We sincerely thank the reviewer for the constructive and thoughtful comments. Below we address the two major concerns in detail and clarify the goals and scope of our theoretical and algorithmic contributions.
>
> **1. On “formalizing a self-evident intuition”**
>
> The reviewer argues that our theorems formally wrap a self-evident intuition. We respectfully disagree and wish to highlight the value of our theoretical framework.
>
> (1). We establish the first principled theory for TIR. While Tool-Integrated Reasoning (TIR) has achieved massive empirical success, the field lacks a formal theory explaining why it succeeds where pure text fails. Our work is not merely an extension of existing literature (e.g., Wu et al., 2025); rather, it stands as the first formal framework dedicated to TIR. It moves the field from empirical observation ("tools work better") to theoretical certainty ("tools enable a strictly larger class of solvable problems").
>
> (2). While the intuition that "tools are efficient" is common, formalizing it is non-trivial and necessary. We offer a precise vocabulary and a structured framework that make previously informal notions analysable. Our intention in the theoretical section is not to claim a deep or surprising theorem, but to provide a formal analytical lens missing in prior work, which enables systematic reasoning about tool-augmented models.
>
> We have clarified the scope of our theorem in the main text (line 116-119).
>
>
> **2. On whether ASPO changes the optimization fixed point or encourages spurious early tool calls**
>
> We thank the reviewer for this excellent question. ASPO is explicitly designed to steer behavior without harming correctness or creating pathological tool usage. We clarify its properties below.
>
> (1). ASPO never rewards incorrect solutions
>
> The auxiliary advantage is applied only to responses that are both correct and contain tool use. A model cannot hack by invoking tools early if the final answer is wrong. The primary objective (solving the problem) remains the hard constraint.
>
> (2). Within the correct-and-tool subgroup, the auxiliary term is zero-sum and bounded
>
> The auxiliary advantage involves normalization p-mean(p). As a result, within the correct-and-tool subgroup, ASPO essentially performs a zero-sum redistribution of advantage (ignoring clipping effects). It does not artificially inflate the total advantage mass for this subgroup. The total advantage mass of "tool-use" or "correct-answer" also remains unchanged. From this perspective, it does not directly encourage more tool calls. Rather, it re-ranks the responses in the correct-and-tool subgroup, expressing a preference for early invocation over late invocation among the correct-and-tool responses.
> In addition, the auxiliary term is clipped to a fraction of the correctness advantage, so the correctness advantage always dominates the optimization target.
>
> (3). Empirical evidence against hacking
>
> Figures 3 and 4(d) show that, for ASPO, overall accuracy does not decrease, and code-pass ratio remains stable. We also manually check a large number of samples and find no trivial or meaningless early calls.
>
> (4). The fixed point problem
>
> We agree with the reviewer that ASPO is not a potential-based shaping method and can't preserve the fix point of policy optimization. But our design goal is different with PBRS: achieving stable behavior steering under a strict correctness-first constraint. The bounded auxiliary term in advantage ensures that ASPO can alter model behavior without altering correctness or introducing instability.
>
> We have revised the manuscript to clarify these points (line 265-267, 277-283).
>
> We thank the reviewer again for the insightful comments, which have helped us improve the clarity and scope of both the theoretical framing and the ASPO design. We have incorporated the clarifications above in the revised version.

---

### Official Review · Reviewer_mLPx · 2025-10-28

**Soundness:** 4
**Presentation:** 3
**Contribution:** 3
**Rating:** 8
**Confidence:** 3

**Summary:**

The paper promotes tool usage (specifically, a Python interpreter) as a general framework for LLM reasoning. It first establishes two theoretical results: (1) Tool-Integrated Reasoning (TIR) can solve a strictly larger class of problems than text-only reasoning, and (2) TIR achieves greater token efficiency in reasoning. On the implementation side, the paper introduces Advantage Shaping Policy Optimization (ASPO), a GRPO-derived reinforcement learning algorithm designed to stabilize TIR training. ASPO incorporates an auxiliary reward that encourages early tool usage by modulating the advantage rather than directly modifying the reward. Experimentally, TIR significantly outperforms text-only RL across standard mathematical benchmarks beyond programming-related tasks. Notably, these gains are not limited to improved arithmetic accuracy, as similar improvements appear in abstract, non-computational problems. Qualitative analysis further reveals distinct reasoning behaviors induced by TIR, including insight-to-computation transformation, exploration and verification through code, and offloading of complex calculations.

**Strengths:**

-	The theoretical analysis in Section 3, while with concerns on practicality as discussed in the Weaknesses, paves motivation to consider TIR as general reasoning framework, especially considering token efficiency.
-	The proposed RL method (ASPO) not only enhances downstream performance but also elicits cognitive patterns essential for establishing TIR as a general reasoning framework rather than a mere computational aid.
-	The experiments are also thoroughly design and conducted not only to verify benefit of TIR and ASPO, but also to rule out null hypotheses via in depth quantitative and qualitative analysis.
-	Personally, I find the algorithmic friendliness wise scoring in Section 4.2 and qualitative study in Section 4.3 revealing various cognitive patterns persuading enough to consider TIR as a general reasoning framework beyond mere calculator assistance.

**Weaknesses:**

-	In Section 3.1.2, the proof of strictness (suggesting that tool augmentation can handle a strictly larger class of problems than text-only LLMs) relies solely on random oracle problems. However, random oracles are not representative of the kinds of problems that LLM-based reasoners are intended to solve. Consequently, it remains uncertain whether the identified strictness region meaningfully extends to practical reasoning or real-world computational tasks.
-	In Section 3.2.2, the proof of strictness (claiming that tool augmentation improves the token efficiency of text-only LLMs for any non-trivial algorithmic problem) assumes that program literals—i.e., code representations—are inherently more concise than the execution traces a text-only LLM must produce to simulate them. While this assumption generally holds, it overlooks the computational cost of executing the tool itself. The reasoning may be reasonable for lightweight tools such as Python interpreters, but it becomes less convincing when the tool involves any resource-intensive process. Because the proof treats all tools equivalently, its notion of token efficiency becomes less meaningful in realistic scenarios.
-	In both Section 3.1.2 and 3.2.2, the proof of inclusion follows from that a tool-augmented model can always choose to not use it. However, that is only true in theory; by incorporating another interface (tool), the model should decide when and not to use it. Without proper selection capacity, the tool-augmented model can and often underperform models without it in certain domains.
-	Minor point: the first and second paragraphs of Section 2 do not connect well; the first covers RL + tool use, while the second explains RL for LLM without tool use.

**Questions:**

-	I am a bit confused on the baseline for experiment in Section 4.1: Is the Qwen3-8B baseline also fine-tuned with the same dataset as TIR? If then, it should be notified otherwise (e.g. Text-Only RL) to differentiate it from vanilla Qwen3-8B.
-	Is there a validation process for Gemini’s aptness as classifier for algorithmic friendliness in Section 4.2? Without validation, we do not know how correct the friendliness score label Gemini generated is.

---

> ### Author Response · Authors · 2025-11-26
> **Response to Reviewer mLPx**
>
> We sincerely thank the reviewer for the positive and insightful assessment, and for the thoughtful comments regarding the theoretical scope and practical considerations. We address each point below.
>
> **Weakness 1: On the use of a random oracle**
>
> We agree that random oracles are idealized. In our work, they serve as a standard existence-style construction to establish strictness. The intention is not to claim realistic tools behave like a random oracle, but to show that there exist deterministic external transitions that a pure-text model cannot emulate with non-negligible probability.
>
> While idealized, the random oracle proof can map directly to realistic scenarios and tools, such as:
>
> - Heavy Computation: The core idea is that the probability of pure-text model generating correct computation results decreases as computational tasks increase in complexity or length. For example, solving a time evolution PDE that requires a numerical solution over N steps. Let the probability of the pure-text model performing a single arithmetic step correctly be constant. Then the probability of generating correct solution decreases exponentially with N. While the TIR model can get correct computational result deterministically with tool.
>
> - Knowledge Retrieval: As discussed in Appendix E, querying an external database for unknown facts (e.g., specific weather data) is mathematically equivalent to a Random Oracle: the pure-text model can only "guess" (prob -> 0), while the tool "retrieves" (prob≈1).
>
> For each of these, we can construct families of problems where the probability of a correct pure-text derivation decays with problem size, while the TIR model retains a deterministic path.
>
> We have revised Section 3.1 and added some paragraphs in appendix (line 166-169, 667-690), to clearly state (1) the role of the random oracle as an existence proof, and (2) how similar strictness arguments apply to practical tools.
>
> **Weakness 2: On the tool cost**
>
> We appreciate the reviewer raising this broader point. Our theoretical analysis focuses on Token Efficiency because the finite context window (token budget) is the fundamental bottleneck for LLMs. Token efficiency therefore captures the expressivity gap that arises purely from the sequential nature of natural language.
>
> In practical scenarios, we can assume a rational agent design: tools are introduced when their cost is far lower than the cost of LLMs generating the equivalent reasoning trace in natural language. This ensures that TIR remains more efficient when accounting for the computational cost of the tool.
>
> We have clarified this distinction and add a paragraph (line 235-240) in the revision.
>
> **Weakness 3: On the proof and scope of theorem**
>
> We fully agree with the reviewer's distinction. Our goal is to study the reachable trajectory space induced by the external tool. The theorem concerns expressible trajectories under fixed model parameters, with the assumption that: the pure-text model and the TIR model share exactly the same underlying language model parameters; the only difference is that the TIR model has access to additional deterministic transitions introduced by the tool. The theorem does not guarantee that a learned policy will effectively choose to do it.
>
> We fully agree that in practical deployment, the model must learn when and how to invoke the tool. However, proving the expansion of the theoretical support is the necessary first step to justify why TIR is a fundamentally more powerful paradigm than pure-text model, distinct from training a better model.
>
> We have revised the theorem statement to emphasize the scope and the key assumption (line 145-149, 157-160).
>
> **Weakness 4**
>
> We thank the reviewer for this point. We have revised the manuscript for a better connect (line 96).
>
> **Question 1: On the pass@k baseline**
>
> We thank the reviewer for pointing this out. The pure-text baseline in Figure 1 is the vanilla Qwen3-8B model without RL fine-tuning. This choice is intentional and conservative: prior work (Yue et al., 2025) reports that RL often decreases pass@k performance at large k. We also evaluated the aime25 pass@k for RL-trained (using the same algorithm and same dataset with TIR model) pure-text model, finding that the vanilla Qwen3-8B model performs better than its RL counterpart in pass@256:
>
> |  | Qwen3-8B (vanilla) | Qwen3-8B (RL) |
> | :--- | :--- | :--- |
> | AIME25 pass@256 | 0.8667 | 0.8333 |
>
> Therefore using the vanilla Qwen3-8B model gives a stronger baseline for evaluating the capability ceiling. We have clarified this baseline choice in the caption of Figure 1 (line 318-322).

---

> > ### Author Response · Authors · 2025-11-26
> >
> > **Question 2: On validation process for algo friendliness**
> >
> > We thank the reviewer for this thoughtful question. We performed several verification checks to ensure the reliability of the "Algorithmic Friendliness" scores:
> >
> > (1) For the lowest-friendliness subset (24 problems with score 1.0–1.5), we manually validated all classifications and found them consistent with the rubric.
> >
> > (2) We query Gemini multiple times per problem and average its outputs to reduce noise.
> >
> > (3) The rubric is detailed and deterministic (Appendix K), which significantly reduces ambiguity in classification.
> >
> > Since the friendliness score is only used for grouping and not for evaluation or supervision, this level of validation is sufficient for robustness. We have clarified validation procedure in the revision (line 385-390).
> >
> > We thank the reviewer again for the constructive and encouraging feedback. We have incorporated the suggested clarifications to further improve the manuscript.

---

> > > ### Comment · Reviewer_mLPx · 2025-11-27
> > >
> > > Thanks for clarifying the scope of the proofs and the accompanying results.
> > > I now agree that the proofs provide meaningful insight into the practical regime of tool-based reasoning, and I have updated my score accordingly.

---

### Official Review · Reviewer_ZcNa · 2025-10-31

**Soundness:** 2
**Presentation:** 3
**Contribution:** 2
**Rating:** 4
**Confidence:** 4

**Summary:**

This paper study how tool use can improve large language model capabilities, both from a theoretical perspective and an empirical one. First, the paper introduces two theorems to characterize how tool-use enhance what LLMs can generate. The first theorem shows that, in general, the support of the distribution of the tool augmented LLM is strictly larger than the one of the text-only LLM (the support of the distribution is defined as the sequences of text with a probability higher than epsilon). The proof rely on using a random oracle as tool, stating that the pure text LLM cannot model the mapping of the random oracle. Then, the authors state a second theorem, showing the superiority of tool augmented LLM under token constraint. This theorem states that under a constraint budget B, and for a non trivial class of problems, there exists a problem size N such that the support of the tool augmented LLM contains one solution of for each problem, while this is not the case for the text only model.

Then, the authors perform an empirical evaluation of tool augmented LLM in the context of reasoning, showing that adding a python interpreter enhance the capabilities of the model for solving math problems. More specifically, the authors performed reinforcement learning algorithm, using the Qwen3-8B model, with and without the use of a python interpreter, on a subset of the DAPO dataset (which comprises math problems). Then, the two models are compared the following benchmarks: AIME24, AIME25 and a subset of Omni-MATH. These experiments show that having access to a python interpreter leads to significantly better results these benchmarks, hinting that it improves the reasoning abilities of the model. Then, the authors explore whether tool augmentation only helps for computation intensive problems or not. To do so, the problems of the Omni-MATH benchmark are splitted into five buckets, based on how much solving these problems rely on computation vs. abstract reasoning. It shows that tool-use improve for all class of problems, and not only for the more computational ones. The last contribution of the paper is a modification of the GRPO algorithm, called ASPO, to elicit the use of tool early in the reasoning. Instead of modifying the reward, which leads to reward hacking, the authors proposed to modify the algorithm itself. They show that with this variant of GRPO, the LLM tends to use the tool more, and earlier in the reasoning.

**Strengths:**

Overall, I am a bit ambivalent about this paper. I believe that the experimental section is strong, and in particular, the results showing that using a python interpreter strongly improve the reasoning abilities of an LLM a interesting (with the small caveat here, that I believe that all the problems considered in the benchmarks have a numerical answer, hence being probably more prone to reasoning with interpreter than problems that have a symbolic answer. In particular, I am wondering whether the tool augmented LLM can guess the answer more often, instead of proving that the answer is correct). I also liked the ablation where authors investigate if tool-use also work for more "abstract" problems vs. computational one. Finally, I think that the modification of the GRPO algorithm is interesting compared to changing the reward, although I am not convinced this is really useful to "force" the model to use the tool more in the setting considered in the article.

**Weaknesses:**

On the other hand, I am really not convinced by the theoretical results, and I believe that they are a big weakness of the paper. Here are my main concerns with these results.

First, I do not believe that the theorems and proofs are "formal" results. For example, some terms are not defined properly, making the theorems vague (eg, what are "non trivial algorithmic problems"?). While I understand the general ideas behind these theorems, I do not believe that they are correct under the current assumptions. For example, to prove that the support of the distribution of the text model in included in the support of the distribution of the tool model, the authors state that

> The tool-integrated model pTIR can generate this same trajectory by adopting a policy of never invoking the external oracle

However, this is not generally true for all tool-augmented models. Some models might actually use the tool every time, depending on how these were trained. More generally, since the distribution is normalized, if some sequences have a higher probability under the tool-augmented model, this means that other sequences have a smaller probability, which likely means that the support of the text only distribution is not included. In particular, it is likely that the probability of "incorrect answers" for certain problems is smaller for the tool augmented LLM, and thus that there is not an inclusion of supp(p_text) in supp(p_tool).

Second, the proof of the first theorem relies on a "random oracle", which is an idealized object that does not exist in practice. So basically, the theorem is only true for a "tool" that actually cannot be instatiated in practice. I think that adding some assumptions on the tool would make the theorem more interesting (but also probably much harder to formalize).

Finally, I also have some concern regarding the significance of the second theorem (about superiority of tool LLM wrt. to text LLM): as currently stated, the theorem assume that calling the tool is computationally free, which is not the case in practice. In particular, it is true that there exists problems for which the solution would require a linear or quadratic solution in terms of number of tokens. But this is also true for the computational complexity of the tool. Thus, I do not believe that only considering the length of the query of the tool is a good way to compare tool augmented LLM with text only ones.

Overall, I do not believe that the theoretical sections add much value to the paper. In particular, I think that the arguments made are informal at best, some of them probably being incorrect as they are currently stated.

**Questions:**

Are all problems considered in the paper have a numerical answer?

Do you have an idea whether sometimes the model "guess" then answer thanks to the interpreter, without actually proving that this is the correct answer?

In Figure 1, both models are trained with RL in the same way, starting from Qwen 3, right? Are there different prompts to elicit tool use for the tool augmented model?

In section 4.4, what is the baseline? Is it GRPO?

---

> ### Author Response · Authors · 2025-11-26
> **Response to Reviewer ZcNa**
>
> We sincerely thank the reviewer for the thoughtful and detailed feedback. We appreciate the reviewer’s positive assessment of our experimental results, the abstract-vs-computational ablation, and the practical motivation behind ASPO. Below we address each concern and clarify the scope and assumptions of our theoretical results.
>
> **1. On the scope and formalism of Theorem 3.1**
>
> We agree that our theoretical analysis should be more explicit about its scope. Our theorem is not about the behavior of a RL-trained policy. Our goal is to study the reachable trajectory space induced by the external tool, not the outcome of RL training. The reviewer is correct that RL training can redistribute probability, but this is outside the scope of our analysis. The theorem concerns expressible trajectories under fixed model parameters, and is therefore unaffected by how training may shift probabilities.
>
> To avoid ambiguity, we will explicitly state the key assumption used in the theorem: the pure-text model and the TIR model share exactly the same underlying language model parameters; the only difference is that the TIR model has access to additional deterministic transitions introduced by the tool. Under this assumption, any text-only trajectory with probability ≥ \epsilon remains reachable in the TIR model because the underlying LM behavior is unchanged.
>
> We have revised the theorem statement to emphasize the scope and the key assumption (line 145-149, 157-160).
>
>
> **2. On the use of a random oracle in the strictness proof**
>
> The reviewer is correct that random oracles are idealized. Our use of a random oracle follows standard practice in theoretical computer science and cryptography to establish existence of strict separations. The goal is not to claim that real-world tools behave as random oracles, but to demonstrate that there exist deterministic tools for which no pure-text next-token model can assign above-ε probability to the target trajectory.
>
> While idealized, the random oracle proof can map directly to realistic scenarios and tools, such as:
>
> - Heavy Computation: The core idea is that the probability of pure-text model generating correct computation results decreases as computational tasks increase in complexity or length. For example, solving a time evolution PDE that requires a numerical solution over N steps. Let the probability of the pure-text model performing a single arithmetic step correctly be constant. Then the probability of generating correct solution decreases exponentially with N. While the TIR model can get correct computational result deterministically with tool.
>
> - Knowledge Retrieval: As discussed in Appendix E, querying an external database for unknown facts (e.g., specific weather data) is mathematically equivalent to a Random Oracle: the pure-text model can only "guess" (prob -> 0), while the tool "retrieves" (prob≈1).
>
> For each of these, we can construct families of problems where the probability of a correct pure-text derivation decays with problem size, while the TIR model retains a deterministic path.
>
> We have revised Section 3.1 and added some paragraphs in appendix (line 166-169, 667-690), to clearly state (1) the role of the random oracle as an existence proof, and (2) how similar strictness arguments apply to practical tools.
>
> **3. On tool computational cost and the meaning of token-efficiency**
>
> We thank the reviewer for raising this important point. We agree that tools have computational costs. However, our theoretical analysis focuses on Token Efficiency because the Context Window (Token Budget) is the fundamental Hard Constraint on LLM feasibility. The theoretical claim is therefore scoped to: Given a finite context window B, can a model express the necessary reasoning trajectory within B tokens? Computational cost of the external tool is orthogonal to this question.
>
> Even when accounting for the computational cost of the tool, TIR remains more efficient because natural language is an inherently flat and sequential execution medium. Any structured computation must be explicitly “unrolled” into a linear token sequence: every intermediate value, loop iteration, branch, and transformation must be verbalized as text. This forces the model to externalize the entire execution trace token-by-token. And pure-text model can't really maintain data structures; you have to reprint, format, and re-expand every time you do something. In contrast, external tools (Python, SymPy, numerical solvers, search engines, verifiers) operate in a fundamentally different execution model: they maintain structured internal state (arrays, hash tables, trees), perform operations through compiled and optimized algorithms, and execute large numbers of internal steps silently without emitting tokens. This difference in execution models (pure-text as a descriptive trace vs. tools as structured stateful computation) ensures that TIR remains more efficient.

---

> > ### Author Response · Authors · 2025-11-26
> >
> > As an empirical note consistent with this analysis, in our experiments the Python tool executed computations faster and with significantly lower computational resource requirements (using CPU rather than GPU) compared to generating the equivalent hundreds of tokens to simulate the calculation.
> >
> > We have clarified this distinction and add a paragraph (line 235-240) in the revision.
> >
> > **4. Responses to Questions**
> >
> > - **Do all evaluated problems have numerical answers?**
> >
> > No. In Omni-MATH-512, many problems' final answers are symbolic expressions, not numbers. Furthermore, our "Algorithmic Friendliness" analysis confirms TIR helps even on abstract, low-computation problems. We have clarified this in the experimental setup section (line 897-901).
> >
> > - **Does the model "guess" the answer?**
> >
> > This is an insightful question. Our case analysis suggests:
> >
> > (1). The Python interpreter is deterministic and cannot directly output the correct final answer without the model generating a correct code.
> >
> > (2). Occasional "lucky guesses" (getting the right answer with wrong/incomplete reasoning) is a known phenomenon in LLMs due to their stochastic nature. However, we observed this behavior in both the pure-text and the TIR model. It is not unique to TIR.
> >
> > (3). Given the large solution space of the benchmarks (e.g., AIME answers are integers 000-999, Omni-MATH involves complex symbolic expressions), the probability of blindly guessing the correct answer is negligible. The significant improvement in pass@k cannot be explained by guess.
> >
> > (4). Crucially, what might look like "guessing" with a tool is often the "Exploration and Verification" pattern (Pattern 2 in Sec 4.3). The model may formulate a hypothesis (a "guess") and use the interpreter to verify it. Unlike blind hallucination in pure text, this is a valid, scientifically sound reasoning strategy enabled by TIR.
> >
> > - **Training setup and different prompts of two models**
> >
> > Yes. Both use Qwen3-8B and the same RL pipeline (DAPO, a GRPO variant), dataset, and hyperparameters. The only difference is that the TIR model receives an additional system prompt defining the tool usage format. We have clarified this in the experimental setup section (line 905).
> >
> > - **Baseline in Section 4.4**
> >
> > The baseline is the standard DAPO (a GRPO variant) with only final answer reward, without the ASPO modification. We have clarified this in the caption of Figure 3 (line 445).
> >
> > We thank the reviewer again for the insightful comments, which have helped us improve the clarity and scope of the theoretical section. We have incorporated the clarifications above in the revised version.

---

### Official Review · Reviewer_nNhn · 2025-11-01

**Soundness:** 3
**Presentation:** 3
**Contribution:** 3
**Rating:** 8
**Confidence:** 3

**Summary:**

This paper provides the first formal proof that Tool-Integrated Reasoning (TIR) strictly expands a Large Language Model's (LLM) capabilities, breaking the constraints of pure-text models. The authors prove that tools expand both the empirical support (unlocking trajectories with previously negligible or zero probability) and the feasible support (making complex algorithms practically achievable via "token efficiency," which would be intractably verbose in natural language).

The paper also introduces Advantage Shaping Policy Optimization (ASPO), a novel algorithm that stably guides model behavior by directly modifying the advantage function, overcoming the instability of traditional reward-based methods. Using the pass@k metric on mathematical benchmarks, the TIR model's performance improvement is evaluated against a pure-text variant. Further deep dives show TIR's benefits are universal, extending even to abstract problems , and identify emergent cognitive patterns of tool usage.

**Strengths:**

ASPO, a new approach is proposed to handle the practical limitations of existing methods for promoting tool usage in the model.

Provides the first formal proof explaining why Tool-Integrated Reasoning (TIR) works, demonstrating it strictly expands an LLM's "empirical" and "feasible" support to overcome the "invisible leash" of pure-text models.

Advocates for a paradigm shift in viewing LLMs: not as monolithic problem-solvers, but as core reasoning engines that intelligently delegate computational tasks to efficient, specialized tools.

Categorizing the problems and demonstrating minimal 'Capability Shrinkage' certainly improves the confidence in the results and a higher performance of the TIR model for problems with high friendliness makes this a very motivating solution. Demonstrating the changes in the model's behaviour as a result of ASPO under sec 4.3 puts the approach into perspective.

**Weaknesses:**

The paper's experiments, while strong, are confined to a single tool (Python interpreter), a single problem domain (mathematical reasoning), and a single base model (Qwen3-8B). This limited scope means the conclusions about the universal benefits of TIR, its generalizability to other tools (like search engines) or domains, and the robustness of the ASPO algorithm across different model families and scales are not fully demonstrated. Authors may want to explicitly state about this limitation, and how they argue that their conclusion is still has external validity of the results.

The paper's own analysis reveals a 1.8% "Capability Shrinkage," where the pure-text model solved problems that the TIR model could not (Figure 2). Will be really good if the authors can provide examples or qualitative analysis of why the TIR model failed on these problems. This will help understand the new failure modes better and appreciate the results better.

Sec L in the appendix indicates that the improvement in the performance is much higher for higher friendliness indicating that the support expansion may not necessarily translate into higher performance. Authors may want to explain this in the draft.

**Questions:**

Can we explore how the 'temperature' of the model affects the improvement in performance? Can authors mention something about this in the draft?

---

> ### Author Response · Authors · 2025-11-26
> **Response to Reviewer nNhn**
>
> We thank the reviewer for the very positive and encouraging evaluation. Below we address the weaknesses and questions.
>
> **Weakness 1: Scope limitation of experiment**
>
> We agree that our experiments use a single tool (Python), one model (Qwen3-8B), and mathematical reasoning tasks. To avoid overclaiming, we add a paragraph (line 911-917) in section "experimental setup" to clarify that:
>
> (1). Our goal is to isolate and interpret the mechanisms of TIR, rather than conduct broad benchmarking.
>
> (2). Python is a representative tool with deterministic, high–information-density behavior well suited to analyzing support expansion and feasible-support expansion.
>
> (3). Appendix E already discusses how our formal arguments extend to other classes of tools (e.g., verifiers, search, memory), even though we do not evaluate them empirically.
>
> **Weakness 2: Analysis of the 1.8% "Capability Shrinkage"**
>
> We thank the reviewer for this excellent suggestion. We analyzed all 9/512 problems where the pure-text model solved the problem but the TIR model did not, and found:
>
> (1). For 6 problems, the pure-text model's success rate was extremely marginal (1/256 or 2/256) and dropped to 0/256 in TIR. This is fully explainable by Monte-Carlo variance, and does not reflect capability shrinkage.
>
> (2). The remaining 3 problems exhibit larger drops. To diagnose this, we evaluated a non-RL TIR model (vanilla Qwen3-8B + Python tool, no RL fine-tuning). This model preserved the pure-text model's accuracy on all 3 problems, indicating that the drops were introduced by RL, not by the TIR mechanism itself. This aligns with prior observations that RL can occasionally induce forgetting on certain items (Yue et al., 2025; Wu et al., 2025).
>
> We have added a section to appendix (Appendix L. ANALYSIS OF THE 1.8% CAPABILITY SHRINKAGE, line 997-1017) to show this.
>
> **Weakness 3: Support expansion and performance gain**
>
> We thank the reviewer for raising this point. We clarify this observation is consistent with our theory. Theorem 3.1 states that "there exists a family of problem instances" for which support expansion is strict. So the support expansion is problem-specific, it does not claim a uniform improvement for every task. Problems with higher algo friendliness benefit more from Python tool. So they are more likely to be in the "support expansion family of problems", result in a higher improvement in the performance. Additionally, even the lowest-friendliness bucket still shows a 9% pass@256 improvement, confirming that TIR provides benefits even for less computational problems.
>
> **Question 1: Effect of Temperature**
>
> We appreciate the reviewer raising this point. To address this, we evaluated both models on AIME25 with temperatures 0.6 and 1.0, using pass@k up to k=256. We found:
>
> (1) For small k, both models show almost identical pass@k curves across temperatures.
>
> (2) For large k, increasing the temperature to 1.0 increases the diversity of TIR trajectories and slightly improves its tail-end pass@k. In contrast, the pure-text model becomes noticeably noisier under temperature 1.0, resulting in lower pass@256. This phenomenon is consistent with prior observations in Yue et al. (2025), which reports that the base model drops but the RL-trained model remains stable when temperature exceeds 1.
>
> (3) So the capability expansion induced by TIR is stable across temperatures, indicating that TIR’s advantage is not due to sampling stochasticity but to its structural support expansion.
>
> We have added a section in appendix (Appendix J. TEMPERATURE SENSITIVITY ANALYSIS, line 938-967) to summarize this observation, with a figure showing the additional inference experiment.
>
> We thank the reviewer again for the encouraging and insightful comments, which have helped us improve the clarity and scope of the manuscript. We have incorporated the clarifications above in the revised version.

---

### Author Response · Authors · 2025-12-02
**Summary of Rebuttal**

We would like to provide a short summary of the reviews and our responses, to help AC quickly assess how the main concerns have been addressed.

## **I. Main Concerns**

All clarifications summarized below have been incorporated into the revised manuscript for the AC’s convenience.

### **1. Theoretical Validity (R2, R3, R4)**

The main concerns raised by R2, R3, and R4 relate to the scope and assumptions of our theoretical framework. Notably, R3, who initially shared similar concerns as R2, was fully convinced by our clarifications and raised the score accordingly.

**(1) Scope and assumptions of the theory**

R2/R3 questioned the statement in proof that The TIR model can reproduce any pure-text model's trajectory.

We clarified that our theoretical analysis is about the reachable trajectory space under fixed model parameters, not about the outcome of RL-training. The key assumption is: the pure-text model and the TIR model share exactly the same underlying language-model parameters; the only difference is the TIR model can use external tools. Under this assumption, TIR model can reproduce any pure-text trajectory with exactly the same probability.

**(2) Random oracles and practical relevance**

R2/R3 noted the use of "random oracle" in the proof is idealized.

We clarified that they serve as a standard existence-style construction to prove strictness, and we show how the argument maps to realistic families such as heavy numerical computation and knowledge retrieval.

**(3) Token efficiency and tool cost**

R2/R3 asked whether feasible-support arguments ignore tool computation cost.

We clarified that our theory focuses on Token Budget because the context window is the fundamental hard constraint for LLMs. And even accounting for tool costs, TIR remains more efficient.

**(4) "self-evident intuition"**

R4 suggested the theory merely formalizes an obvious intuition.

We clarified that: (1) This is the first principled theoretical framework dedicated to TIR, filling a gap in the literature; (2) Turning intuition (“tools are efficient”) into precise, analyzable definitions is non-trivial and necessary for systematic reasoning.


### **2. Algorithmic Stability of ASPO (R4)**

R4 questioned whether ASPO may change the optimization fixed point or induce spurious early tool calls.

We clarified that our goal differs from PBRS: ASPO aims for stable behavior steering under a strict correctness-first constraint.The auxiliary advantage is applied only to correct-and-tool trajectories and acts as a zero-sum redistribution within that subgroup, which does not directly encourage more tool calls. Empirical evidence also shows no reward hacking or spurious early calls.

### **3. Experimental Concerns (R1, R2, R3)**

**(1) Limited experimental scope (R1)**

We added text clarifying that Python serves as a representative deterministic tool to isolate TIR mechanisms rather than to perform broad benchmarking.

**(2) 1.8% “capability shrinkage” (R1)"**

We added a new appendix section showing that this small shrinkage is due to sampling variance and RL training, rather than a limitation of TIR.

**(3) Temperature sensitivity (R1)**

We evaluated pass@k on AIME25 at different temperatures and added a new appendix section showing TIR’s gains are stable across temperatures.

**(4) Answer type and guessing (R2)**

We clarified that many problems have symbolic answers, and the large pass@k gains cannot be explained by guessing.

**(5) Baseline choice for pass@k (R3)**

We clarified that the baseline is vanilla Qwen3-8B (not RL), which is a stronger baseline for pass@k.

**(6) Validation of algorithmic friendliness (R3)**

We described our verification steps: manual checking of the lowest-friendliness bucket, multiple-query averaging, and a detailed rubric.

## **II. Status after Discussion**

- R3 explicitly stated that the clarifications resolved their concerns and raised the score at 01:24 AM UTC on Nov 27, before the OpenReview bug.

- R1, R2, R4 have not replied further.

We sincerely thank the AC and all reviewers for their time, effort, and constructive feedback, which substantially improved our paper.

---

### Meta-Review · Area_Chair_tbKN · 2026-01-06

**Summary:**

In this submission, titled "Understanding Tool-Integrated Reasoning", the authors make the case for thinking of tool-use as a key component that broadens the usability and generalizability of reasoning LLMs. The submission argues this through both a theoretical formulation, and a number of experimental results.

**Reviewer Concerns:**

Overall there is agreement that the paper is helpful, and the while experimental results are confined to use of a Python tool on mathematical problems, which is a bit limiting compared to the submission's ambition to "advocate for a paradigm shift: viewing LLMs not as monolithic problemsolvers, but as core reasoning engines that intelligently delegate computational tasks to specialized,
efficient tools", they are interesting.

The main concern raised by the reviewers is with the theoretical formalization of tool use and the proof tool (randomized oracles) used to investigate the benefits of tool use. The key part of the revised proof in Appendix C is, rephrased by me, that an argument that for queries to the model that involve a hash function with n bits require a model without access to this hash 2**n guesses to evaluate the function, whereas a model with access to the hash function requires only one query. The proof then argues that
> For any non-negligible probability pTIR(y∗|x) and a sufficiently large m, the corresponding qtext(y∗|x)becomes arbitrarily small. We can therefore always choose an ε such that qtext(y∗|x) < ε ≤pTIR(y∗|x). So we find that y∗ /∈ suppε(qtext)

But, this does not mean that y is not in the support of q_text - actually 1/2**n is a positive term so y is contained in the support, it just becomes less likely as n increases (but crucially never leaves the support).

Aside from the detailed statement of the theorem, the particular formalization chosen here is quite far from practical usage of tools in models, and arguably not a formalization amenable to the claim that "[this] work provides the first formal proof that TIR fundamentally expands an LLM's capabilities".

----
Later, in the detailed proof for Theorem 3.4, the submission argues that
> This follows directly from the divergent scaling properties of natural language versus programmatic representations, as illustrated in Tables 1-4

but tables 1-4 only contain a handful of examples. At no point is this statement formalized sufficiently?

---
These concerns echo why reviewers are concerned and do not "believe that the theorems and proofs are "formal" results. For example, some terms are not defined properly, making the theorems vague (eg, what are "non trivial algorithmic problems"?). While I understand the general ideas behind these theorems, I do not believe that they are correct under the current assumptions." and "Although the paper positions its theoretical proofs as a primary contribution (as stated in the Contributions), the core results largely repackage a self-evident intuition: if a deterministic tool can collapse a long derivation into a one-step call, its reachable set will strictly dominate text-only generation. The theorems read more like formal wrapping of an intuition than analysis that meaningfully narrows uncertainty about real systems".

Reviewers also raise further issues with the proofs in the appendix, such as that "Section 3.1.2 and 3.2.2, the proof of inclusion follows from that a tool-augmented model can always choose to not use it. However, that is only true in theory; by incorporating another interface (tool), the model should decide when and not to use it. "

---

After careful reading of the proofs in the appendix of the revised version and the authors' general statement I am still not convinced that these statements are an accurate or appropriate characterization of the problem.

I think this is a bit unfortunate, as I do think the experimental evaluation of math, and the modifications to GRPO are interesting conceptual contributions, that are being sunk here with the focus on proving that tool use expands the support of the distribution, have some merit, and a revised version focusing on these results would be a good submission to a future conference.

**Reviewer Scores:**

ZcNa no change, nNhn no change, mLPx noted that they would increase their score, but I do not find that the response addresses their stated concerns. zSw2 no change projected.

---

### Decision · Program_Chairs · 2026-01-26

Reject